# SFP: State-free Priors for Exploration in Off-Policy Reinforcement Learning

**Marco Bagatella**  *marco.bagatella@inf.ethz.ch*
*Department of Computer Science*
*ETH Zürich*

**Sammy Christen**  *sammy.christen@inf.ethz.ch*
*Department of Computer Science*
*ETH Zürich*

**Otmar Hilliges**  *otmar.hilliges@inf.ethz.ch*
*Department of Computer Science*
*ETH Zürich*

**Reviewed on OpenReview:** *https://openreview.net/forum?id=qYNfwFCX9a*

## Abstract

Efficient exploration is a crucial challenge in deep reinforcement learning. Several methods, such as behavioral priors, are able to leverage offline data in order to efficiently accelerate reinforcement learning on complex tasks. However, if the task at hand deviates excessively from the demonstrated task, the effectiveness of such methods is limited. In our work, we propose to learn features from offline data that are shared by a more diverse range of tasks, such as correlation between actions and directedness. Therefore, we introduce state-free priors, which directly model temporal consistency in demonstrated trajectories, and are capable of driving exploration in complex tasks, even when trained on data collected on simpler tasks. Furthermore, we introduce a novel integration scheme for action priors in off-policy reinforcement learning by dynamically sampling actions from a probabilistic mixture of policy and action prior. We compare our approach against strong baselines and provide empirical evidence that it can accelerate reinforcement learning in long-horizon continuous control tasks under sparse reward settings.

## 1 Introduction

Exploration is a fundamental issue in reinforcement learning (RL): in order for an agent to maximize its reward signal, it needs to adequately cover its state space and observe the outcome of its actions. This becomes increasingly difficult when dealing with large, continuous state and action spaces, as is the case in many real world applications. Despite a large and fruitful body of research on exploration (Bellemare et al., 2016; Osband et al., 2016; Tang et al., 2017; Osband et al., 2018; Azizzadenesheli et al., 2018; Burda et al., 2019; Dabney et al., 2021; Ecoffet et al., 2021; Vulin et al., 2021), most general-purpose algorithms remain based on $\epsilon$-greedy exploration (Mnih et al., 2015) or entropy-regularized Gaussian policies (Haarnoja et al., 2018). In the absence of an informative reward signal, both methods rely on uniformly sampling actions from the action space, independently of the history of the agent. Unfortunately, in sparse reward settings, achieving positive returns by uncorrelated exploration becomes exponentially less likely as the horizon length increases (Dabney et al., 2021).

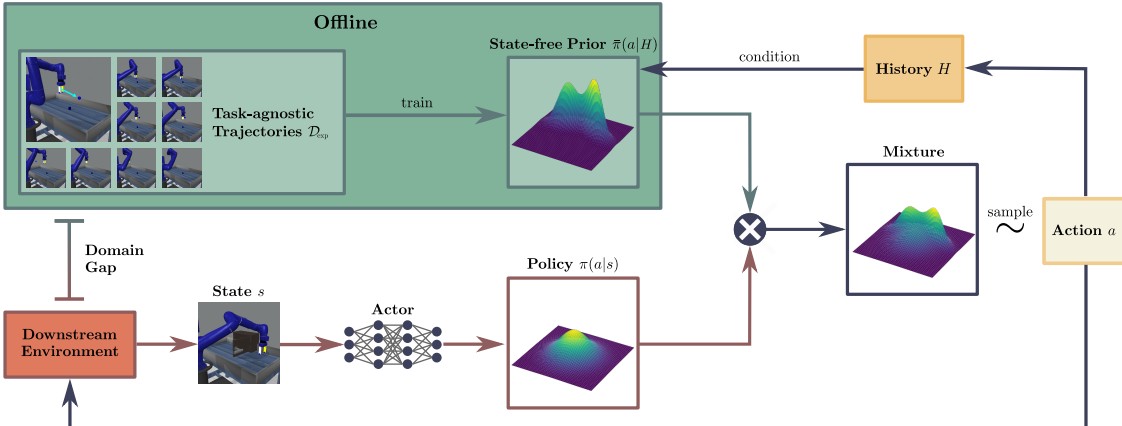

**Figure 1:** SFP: A state-free prior is trained on weakly informative trajectories, such as reaching random goals in a reaching task (top left). Actions are then sampled from a dynamic mixture between the state-free prior and the policy in downstream learning of more complex tasks. Our method works with both vector-based and image-based state inputs.

A promising approach to achieve efficient exploration is that of using a behavioral prior to guide the policy (Pertsch et al., 2020; Tirumala et al., 2020; Singh et al., 2021). Typically, this is learned from expert trajectories as a state-conditional action distribution. Behavioral priors are able to foster directed exploration (Singh et al., 2021), by assuming a strong similarity between the agent and expert tasks. However, an agent should ideally be able to produce efficient explorative behaviors across a broad range of diverse tasks, even if they are rather different from those demonstrated (Parisi et al., 2021). Similarly, it should be possible to leverage information collected on simpler tasks to enable learning in more complex scenarios (Florensa et al., 2017; Gehring et al., 2021).

Let us, for instance, consider a surveying robot that was trained from scratch in a simple room, and needs to be redeployed to a complex construction site: even if observations in the second environments are drastically different from those previously received (e.g. due to different lighting conditions and terrain patterns), relevant information could still be recovered from past experiences. We argue that such information often includes correlation and directedness in the robot's behavior. In particular, these characteristics of the robot behavior are not necessarily related in their entirety to the state space it operates in.

In this paper, we thus propose to focus on the temporal structure of demonstrations rather than on task-specific strategies. We find that this choice enables knowledge transfer from simple offline trajectories to a diverse family of more complex tasks. Our method, which we dub **S**tate-**f**ree **P**riors for exploration in Off-policy Reinforcement Learning (SFP), introduces state-free priors as *state-independent non-Markovian action distributions* $\bar{\pi}(a_t|a_0^{t-1})$, modeling promising actions conditioned on past actions. We find that this class of priors is sufficient for capturing desirable properties for exploration, such as directedness and temporal correlation. This information is often learnable from few expert trajectories collected in simple environments (e.g. reaching uniformly sampled end-effector positions for a robotic arm, see Figure 1). While we assume that such trajectories display qualities such as correlation and directedness, and in particular reflect good exploration strategies for the task at hand, we do not require them to explicitly encode state-action dependencies specific to downstream tasks. State-free priors can then be used to guide exploration in more complex tasks, in which observed states are not guaranteed to belong to the expert state distribution, and behavioral priors are, as a result, inherently limited.

Furthermore, we propose a principled manner of integrating action priors into the Soft Actor Critic framework (Haarnoja et al., 2018). In downstream learning of more complex tasks, our method samples actions from a dynamic mixture between the policy and state-free prior. Crucially, we derive an update rule for learning the mixing coefficient by directly optimizing a (max-entropy) RL objective. Our dynamic integration allows the policy to receive strong guidance when needed, while crucially retaining the ability to explore diverse actions when the prior is no longer beneficial.

In our experiments, we first analyze the choice of conditioning variables for action priors by measuring their effectiveness in accelerating downstream RL. We then focus on state-free priors and verify their capability to produce correlated and directed behavior. Most importantly, we provide empirical evidence that SFP can leverage weakly informative trajectories to accelerate learning. For this purpose, our method is evaluated

against several baselines in complex long-horizon control tasks with sparse rewards. Finally, we present further applications of SFP, including leveraging offline state-based trajectories to (a) accelerate visual RL or (b) guide exploration when the task is corrupted by biased observations.

Our contributions can be organized as follows:

1. We propose learnable non-Markovian action priors conditioned on past actions. We show that sampling from these priors produces directed and correlated trajectories.

2. We introduce a principled manner of integrating action priors into the Soft Actor Critic framework (Haarnoja et al., 2018).

3. We show how state-free priors can be learned from few expert trajectories on simple tasks (e.g. reaching) and used to improve exploration efficiency in more complex, unseen tasks (e.g. opening a window), even across fundamentally different settings (e.g., from non-visual to visual RL).

After discussing our method's novelty and related literature in Section 2, we introduce our setting in Section 3. The method is described in Section 4, while empirical evidence of its effectiveness is reported in Section 5. Finally, Section 6 contains a brief closing discussion of our work. Code and video are available on our project page[1].

## 2   Related Work

While we limit this section to essential topics, further discussion can be found in Appendix I.

**Temporally-Extended Exploration**   Several works have attempted to directly address the inability of traditional methods, such as $\epsilon$-greedy or uniform action sampling (Lillicrap et al., 2016; Haarnoja et al., 2018), to produce correlated trajectories. A recent study (Dabney et al., 2021) highlights this issue and shows how repeating random actions for multiple steps is sufficient to significantly accelerate Rainbow (Hessel et al., 2018) on the Atari benchmark (Bellemare et al., 2013). Similarly, Amin et al. (2021) propose a non-learned policy inspired by the theory of freely-rotating chains in polymer physics to collect initial explorative trajectories in continuous control tasks. Both methods pinpoint a fundamental issue, but rely on scripted policies which are hand-crafted for a particular family of environments. On the other hand, our prior is learned, and does not require engineering an explorer, which can be unfeasible for complex tasks. A similar approach was previously proposed by Bogdanovic & Righetti (2019), who also leverage simple trajectories to accelerate learning in new tasks by designing a learned exploration model (LEP) conditioned on a sequence of recent states. Our method further explores this direction by removing the assumption of a shared state space between tasks, and enables knowledge transfer to a wider range of tasks. Moreover, compared to previous works, our integration of priors into off-policy RL is not limited to initial trajectories and dynamically adjusts the likelihood of sampling actions from the prior.

**Hierarchical Reinforcement Learning**   Another approach to tackle exploration-hard tasks is to rely on a hierarchical decomposition into different levels of *temporal* and *functional* abstraction (Parr & Russell, 1998; Dietterich, 2000; Sutton et al., 1999; Dayan & Hinton, 2000). For instance, tasks can be decomposed into high level planning and a set of low-level policies, often referred to as *skills* (Konidaris & Barto, 2007; Eysenbach et al., 2019) or *options* (Sutton et al., 1999; Bacon et al., 2017). This effectively reduces the planning horizon and allows efficient solving of complex tasks (Bacon et al., 2017; Vezhnevets et al., 2017; Nachum et al., 2018; Levy et al., 2019; Christen et al., 2021). In general, such approaches only target temporal correlation *within* skills, as each skill is generally selected independently from the last one. Incidentally, our method is not designed to achieve temporal abstraction, but can be interpreted in a hierarchical framework (Schäfer et al., 2021) in which a high-level criterion (the mixing function) governs a probabilistic choice between an explorer (state-free prior) and an exploiter (policy). Our method's application is also inherently related to the works of Florensa et al. (2017); Gehring et al. (2021), which propose to extract skills on simple training tasks and

---

[1] https://eth-ait.github.io/sfp/

deploy them in more complex scenarios. In contrast with these methods, training a state-free prior does not require access to the training environment, but only to offline trajectories.

**Behavioral Priors** Behavioral priors generally represent state-conditional action distributions, modeling promising actions for the current state of the environment. Such priors can be learned jointly with the policy in the context of KL-regularized RL (Tirumala et al., 2019; 2020). An important line of work deals with information asymmetry (Galashov et al., 2019; Salter et al., 2022), which consists in restricting the information available to the prior. As a result, learned priors can more easily generalize to different settings, or be trained jointly to accelerate policy learning. In this context, Tirumala et al. (2020) briefly mention the possibility to condition behavioral priors exclusively on a vector of previous actions. Our work can be seen as a broad empirical exploration in this direction, coupled with an integration scheme that is better suited to this class of priors. A second approach consists in learning behavioral priors from expert policies on related tasks. This is the case for several works (Peng et al., 2019; Pertsch et al., 2020; 2021; Ajay et al., 2021) which adopt a Gaussian behavioral prior in a latent skill-space. In particular, Pertsch et al. (2020) report that a prior is crucial to guiding a high-level actor in an HRL framework. An important contribution to the field is made by PARROT (Singh et al., 2021), which focuses on a visual setup and introduces a flow-based transformation of the action space to allow arbitrarily complex prior distributions. We extend this idea to prior action distributions that are not conditioned on the current state or a part thereof, but rather on past actions, and are therefore non-Markovian. Moreover, we propose a novel, more flexible integration of the prior distribution into the learning algorithm. Finally, we overcome the reliance on a strong similarity between states observed by the expert and the agent.

## 3 Background

### 3.1 Setting

Reinforcement learning (RL) is the problem that an agent faces when learning to interact with a dynamic environment. Albeit with a slightly different definition, we formalize the environment as a goal-conditioned Markov Decision Process (gc-MDP) (Nasiriany et al., 2019), that consists of a 6-tuple $(\mathcal{S}, \mathcal{A}, \mathcal{G}, \mathcal{R}, \mathcal{T}, \gamma)$, where $\mathcal{S}$ is the state space, $\mathcal{A}$ is the action space, $\mathcal{G} \subseteq \mathcal{S}$ is the goal space, $\mathcal{R} : \mathcal{S} \times \mathcal{G} \to \mathbb{R}$ is a scalar reward function, $\mathcal{T} : \mathcal{S} \times \mathcal{A} \to \Pi(\mathcal{S})$ a probabilistic transition function that maps state-action pairs to distributions over $\mathcal{S}$ and, finally, $\gamma$ is a discount factor. Assuming goals to be drawn from a distribution $p_{\mathcal{G}}$, the objective of an RL agent can then be expressed over a time horizon $T$ as finding a probabilistic policy $\pi^{\star} = \arg\max_{\pi} \mathbb{E}_{g \sim p_{\mathcal{G}}} \sum_{t=0}^{t=T} \gamma^t \mathcal{R}(s_t, g)$, with $s_t \sim \mathcal{T}(s_{t-1}, a_{t-1})$ and $a_{t-1} \sim \pi(s_{t-1}, g)$. In order to simplify notation, from this point on, we will implicitly include the goal into the state at each time step: $s_t \leftarrow (s_t, g)$.

We focus on long-horizon control problems with continuous state and action spaces and sparse rewards, i.e., non-zero only after task completion. Although our method can be generally applied to off-policy RL methods, we build upon Soft Actor Critic (Haarnoja et al., 2018), due to its wide adoption in these settings.

Finally, in contrast with several behavioral prior approaches (Galashov et al., 2019; Pertsch et al., 2020; Singh et al., 2021), we adopt a more general and challenging setting. First, we do not assume that prior information on the structure of the state space is available. Second, while we also assume access to a collection of trajectories $D_{\exp} = \{(s_0^i, a_0^i, s_1^i, a_1^i, \ldots, s_N^i, a_N^i)\}_{i=0}^{L}$, we do not require trajectories to be collected on a task that is similar to the one at hand. From this point on, we refer to the task used for collecting data as the *training task*, and to the task the RL agent has to complete as the *downstream task*.

### 3.2 Behavioral Priors

A behavioral prior $\bar{\pi}(a|s)$ (Pertsch et al., 2020; Singh et al., 2021) is a state-conditional probability distribution over the action space (cf. Figure 2a). Behavioral priors can be trained to assign high probability to *useful* actions with respect to the current state, and hence be used to accelerate RL. A behavioral prior can only guide the policy effectively as long as prior information on the structure of the state space is available (Galashov et al., 2019), or the prior has been trained on data collected on a closely related task (Pertsch et al., 2020; Singh et al., 2021).

In our settings, the structure of observations is unknown and expert trajectories $D_{\text{exp}}$ are weakly informative or collected on simple tasks. As a consequence, the distribution of training states might not match the distribution of states produced by downstream tasks: behavioral priors will then be evaluated on out-of-distribution samples and their performance will degrade drastically, as shown in Section 5.1.

## 4 Method

Our method relies on the integration of a learned state-free prior into an off-policy RL algorithm. Thus, we first define and discuss the class of action priors of interest, and later describe how they can be integrated into existing off-policy algorithms.

### 4.1 State-free Priors

Within the settings outlined in Section 3.1, it is still possible to extract and transfer knowledge from the expert dataset $D_{\text{exp}}$ to the agent. In this case, conditioning an action prior on the current state might be both insufficient and counterproductive, as a state-conditional action prior would receive out-of-distribution samples as inputs, due to the distribution shift between states from the training and downstream task. For this reason, we shift our focus towards modeling the temporal correlation of expert trajectories.

In order to recover this information, we thus propose to learn a *state-free prior*. A state-free prior is a non-Markovian, state-independent action prior, representing a probability distribution over the action space that is conditioned on past actions: $\bar{\pi}(a_t|a_0^{t-1})$ (see Figure 2b, c). By renouncing the informativeness of states with respect to expert actions, state-free priors retain the ability to model temporal correlation, which crucially enables knowledge transfer to tasks with a fundamentally different state distribution with respect to demonstrations. This is indeed a viable strategy in a hard-exploration setting, when no prior information is available on the state space and no reward is observed: in this case, extracting any information from the state remains challenging. Our empirical evidence suggests that the simplest form of state-free priors, i.e., $\bar{\pi}(a_t|a_{t-1})$ (see Figure 2c) is surprisingly competitive with variants conditioned on multiple past actions (cf. Section 5.1) and is sufficient to capture non-trivial temporal relations.

Independently of the conditioning variables, state-free priors can conveniently be modeled as conditional generative models and trained through empirical risk minimization. For the purpose of this paper, we choose to use the conditional variant of the Real Non Value Preserving Flow (Dinh et al., 2017; Ardizzone et al., 2019), which is well suited for Euclidean action spaces (Singh et al., 2021). Moreover, our integration in the SAC framework allows arbitrarily complex prior distributions, which Real NVP Flows are in principle able to capture.

In the context of Real NVP Flows, training samples are actions $a \in \mathcal{A}$, paired with conditioning variables $a_0^{t-1}$, thus the learned mapping is $a = f_\theta(z; a_0^{t-1})$, with $z \sim \mathcal{N}$. Since $f_\theta$ is invertible, it is possible to analytically compute the likelihood of a single training pair $(a_t, a_0^{t-1})$ and maximize its expected value through standard gradient-based optimization techniques. An empirical justification of this choice is found in Appendix B, while implementation details are reported in Appendix K. For a complete introduction to Real NVP Flows, we refer the reader to Dinh et al. (2017).

One final concern regards the nature of the data for training the state-free prior. The main requirements for the training data are two: (1) the task on which the data is collected needs to share the same action space of the downstream task and (2) the training trajectories should display the desired qualities of correlation and directness. In general, we adopt weakly informative expert trajectories generated by achieving random goals in simple environments. Such simple trajectories can be produced from scratch using standard RL or generated by a scripted policy. We remark that, in contrast with

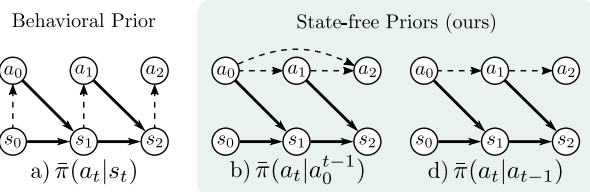

**Figure 2:** Graphical models of different priors: from left to right, a behavioral prior, a general state-free prior and a single-step state-free prior. Solid arrows represent the environment's transition function $\mathcal{T}$, while dashed arrow indicate conditional modeling.

existing approaches (Pertsch et al., 2020; Singh et al., 2021), this framework poses very weak requirements on the similarity between the task used for data collection and the target task. As we show in Section 5.4, this allows our method to transfer knowledge across different tasks and settings, such as from a simple reaching task with access to the true state of the system to a window-closing task in a visual RL setting. Limitations of our method with respect to training data are discussed in Section 6.

## 4.2 Soft Actor Critic with SFP

The main challenge introduced by state-free priors stems from their non-Markovianity, which renders existing integrations schemes unsuitable Singh et al. (2021); Tirumala et al. (2020) (see Section 5.2 for an empirical evaluation and a broader discussion). For this reason, we introduce a novel method to integrate action priors in an off-policy RL framework. While our method can be formally derived for Markovian priors, it can also be generalized to non-Markovian action distributions by allowing further bias in Q-value estimates. In the following derivations, we will consider a general state-dependent non-Markovian prior $\bar{\pi}(a_t|H_t)$ with $H_t = (s_0^t, a_0^{t-1})$, but the integration remains applicable to state-free or behavioral priors alike (for briefness, we will interchangeably use the compact notation $\bar{\pi}(a_t)$). The key strategy consists of sampling actions from a dynamically weighted mixture between the policy and a fixed prior distribution. We demonstrate it as an integration into the SAC framework.

SAC is an off-policy actor critic algorithm that trains a stochastic policy $\pi_\phi$ and state-action value function estimator $Q_\theta$ in an off-policy way; its objective is designed to pursue rewards while maximizing the entropy of its policy. When prior knowledge on the structure of the environment or task is available, simply sampling actions from a maximal entropy policy $\pi_\phi$ may not be optimal. On the other hand, blindly sampling from a fixed action prior $\bar{\pi}$ prevents exploitation of reward signals as well as any behavior which is not encoded in the prior. Ideally, it is desirable to control the degree to which actions are sampled from the prior. We propose to achieve this in a natural way by sampling actions from a mixture $\tilde{\pi}$ between the policy $\pi_\phi$ and the prior $\bar{\pi}$:

$$a_t \sim \tilde{\pi}(\cdot|s_t) = (1 - \lambda_t)\pi_\phi(\cdot|s_t) + \lambda_t\bar{\pi}(\cdot), \qquad (1)$$

where the mixing parameter $\lambda_t$ is bounded ($0 \leq \lambda_t \leq 1$) and computed dynamically at each step $t$.

---

**Algorithm 1** SAC with SFP

1: Train action prior $\bar{\pi}(a_t|H_t)$
2: Initialize parameters $\theta, \phi, \omega$
3: **for** each iteration **do**
4:     Observe $s_0$
5:     Initialize history $H_0 = (s_0)$
6:     **for** each environment step **do**
7:         $\lambda_t = \Lambda_\omega(s_t)$
8:         $a_t \sim \tilde{\pi} = (1 - \lambda_t)\pi(a_t|s_t) + \lambda_t\bar{\pi}(a_t|H_t)$
9:         $s_{t+1} \sim \mathcal{T}(s_t, a_t)$
10:        $H_{t+1} = H_t \cup (a_t, s_{t+1})$
11:     **end for**
12:     **for** each gradient step **do**
13:         Update $\theta, \phi, \omega$
14:     **end for**
15: **end for**

---

In principle, it is desirable to bias the mixture $\tilde{\pi}$ towards the prior $\bar{\pi}$ or the policy $\pi$, depending on which of the two is more likely to reach the goal. Interestingly, if the mixing weight is computed through a parameterized function of states $\lambda_t = \Lambda_\omega(s_t)$, this behavior can be naturally recovered by maximizing the maximum entropy objective with respect to the mixture $\tilde{\pi}$:

$$\arg\max_{\pi_\phi, \Lambda_\omega} \mathbb{E}_{\tau \sim \tilde{\pi}} \left[ \sum_{t=0}^{\infty} \gamma^t \left( \mathcal{R}(s_t, a_t) + \alpha\mathcal{H}(\pi_\phi(\cdot|s_t)) \right) \right]. \qquad (2)$$

We note that, while this expectation is computed over trajectories sampled from the mixture $\tilde{\pi}$, the entropy term $\mathcal{H}(\pi_\phi(\cdot|s_t))$ is computed on the policy $\pi_\phi$ alone, as it is not desirable to incentivize exploration of states according to the prior $\bar{\pi}$'s entropy.

Through straightforward derivations (see Appendix A), it is possible to derive learning objectives for the Q-estimator $Q_\theta^{\tilde{\pi}}$ and for the policy $\pi_\phi$, as well as for a mixing function $\Lambda_\omega$. Given a distribution $\mathcal{D}$ of observed transitions, the loss functions for the Q-estimator can be defined as:

$$J_{Q_\theta^{\tilde{\pi}}} = \mathbb{E}_{(s,a,s') \sim \mathcal{D}} \left[ \left( Q_\theta^{\tilde{\pi}}(s, a) - y_t(s, a, s') \right)^2 \right], \qquad (3)$$

where the target for the Q-value is computed as

$$y_t(s, a, s') = \mathcal{R}(s, a) + \gamma \bigg( Q_\theta^{\tilde{\pi}}(s', \tilde{a}') - \alpha \log \pi_\phi(a'|s') \bigg)$$
$$\text{with } \tilde{a}' \sim \tilde{\pi}(\cdot|s'), a' \sim \pi_\phi(\cdot|s'). \tag{4}$$

On the other hand, the policy $\pi_\phi$ and the mixing function $\Lambda_\omega$ can be trained to maximize the value function (Haarnoja et al., 2018) for the mixture $\tilde{\pi}$. The value function can be decomposed in expectations over actions sampled from the prior $\bar{\pi}$ and the policy $\pi_\phi$ as derived in Appendix A:

$$V^{\tilde{\pi}}(s) = \lambda \mathop{\mathbb{E}}_{\bar{a} \sim \bar{\pi}(\cdot)} \bigg[ Q_\theta^{\tilde{\pi}}(s, \bar{a}) \bigg] + (1 - \lambda) \mathop{\mathbb{E}}_{a \sim \pi_\phi(\cdot|s)} \bigg[ Q_\theta^{\tilde{\pi}}(s, a) \bigg] - \alpha \mathop{\mathbb{E}}_{a \sim \pi_\phi(\cdot|s)} \bigg[ \log(\pi_\phi(a|s)) \bigg], \tag{5}$$

where $\lambda = \Lambda_\omega(s)$ is the mixing weight computed on the current state $s$. This decomposed value function can be maximized by minimizing the two following losses with respect to the policy's and the mixing function's parameters:

$$J_{\pi_\phi} = - \mathop{\mathbb{E}}_{(s) \sim \mathcal{D}} \bigg[ \big(1 - \Lambda_\omega(s)\big) \big( Q_\theta^{\tilde{\pi}}(s, a) - \alpha \log \pi_\phi(a|s) \big) \bigg]$$
$$\text{with } a \sim \pi_\phi(\cdot|s), \tag{6}$$

$$J_{\Lambda_\omega} = - \mathop{\mathbb{E}}_{(s) \sim \mathcal{D}} \bigg[ \Lambda_\omega(s) \big( Q_\theta^{\tilde{\pi}}(s, \bar{a}) - Q_\theta^{\tilde{\pi}}(s, a) \big) \bigg]$$
$$\text{with } \bar{a} \sim \bar{\pi}(\cdot), a \sim \pi_\phi(\cdot|s). \tag{7}$$

The three objectives can be empirically estimated and minimized through standard procedures, as reported in Haarnoja et al. (2018) and in Appendix A. We further note that, as expected, Equation 7 encourages high mixing weights $\lambda_t = \Lambda_\omega(s_t)$ in case Q-values for actions sampled from the prior $\bar{\pi}$ are higher than those for actions sampled from the policy $\pi_\phi$, and vice versa (see Appendix F for an empirical analysis on the evolution of $\lambda$ during training). We remark that the resulting mechanism is similar in spirit to the Q-filter presented in Nair et al. (2018), which is however originally only applied as a binary signal for weighting a behavioral cloning loss.

Algorithm 1 summarizes (in blue) the necessary modifications for integrating the action prior into the SAC framework. Namely, actions are sampled from a mixture (line 8) weighted according to the output of a mixing function (line 7). Finally, the history of the agent needs to be initialized (line 5) and updated at each step (line 10). Update rules for $\theta$, $\phi$ and $\omega$ (line 13) are computed by minimizing the objectives in Equations 6, 3 and 7.

## 5 Experiments

We evaluate our method in a series of experiments to empirically validate our contributions. First, in Section 5.1, we compare the effectiveness of different action priors to justify the choice of state-independent conditioning. Then, in Section 5.2 we evaluate our integration against existing ones in the context of state-free priors. In Section 5.3, we verify that sampling from a state-free prior can produce correlated and state-covering behavior, without the need to hand-craft an exploration policy. Next, we show how our method can improve exploration efficiency in unseen long-horizon tasks by comparing against various baselines in state-based RL (Section 5.4). Finally, we present a proof of concept of different applications for state-free priors, i.e., in visual RL and when dealing with biased observations (Section 5.5). Ablations for our method and baselines are provided in Appendix B.

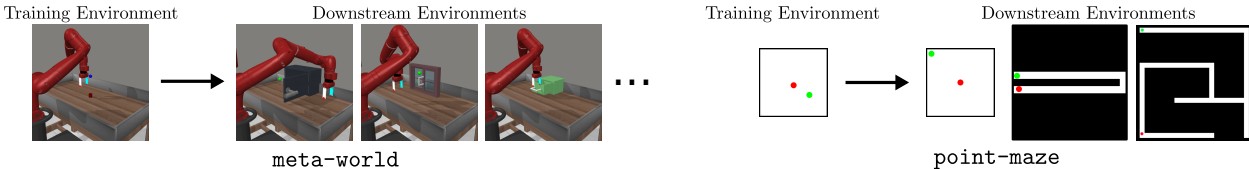

**Figure 3:** Non-exhaustive overview of environments used in our experimental validation.

**Baselines**   We now present a brief introduction to the baselines we consider, while implementation details are provided in Appendix K.3.

- **SAC**: vanilla Soft Actor Critic (Haarnoja et al., 2018).
- **SAC+BC**: SAC with warm-started policy through behavior cloning.
- **SAC-PolyRL**: SAC with locally self-avoiding walks (Amin et al., 2021).
- **PARROT-state**: flow-based behavioral prior enforced through a transformation of the action space (Singh et al., 2021). We benchmark a state-based variant in non-visual settings. We found other methods based on behavioral priors to perform similarly, while being arguably more complex (see Appendix H for an additional baseline).

**Environments**   We evaluate our method on three types of domains, namely robotic manipulation, maze navigation and robotic locomotion:

- `meta-world`:  robot manipulation tasks from the MT10 benchmark in the publicly available `meta-world` suite (Yu et al., 2020).
- `point-maze`: qualitatively different 2D maze structures, navigated by a point-like agent Pitis et al. (2020)
- `gym-mujoco`: widely adopted continuous control environments for locomotion of simulated robots Brockman et al. (2016); Todorov et al. (2012).

The first two suites define diverse tasks that share similar underlying dynamics (e.g. navigating mazes with different structures, or interacting with different objects), and are therefore used to benchmark the ability to transfer knowledge from offline trajectories to complex downstream tasks (Section 5.1, 5.3, 5.4). In this case, offline trajectories are collected in simple *training tasks* that involve reaching uniformly sampled goals in an empty environment (`reach` for `meta-world` and `room` for `point-maze`). RL agents are then trained and evaluated on a wider range of *downstream tasks*, involving object manipulation, such as opening a window, or navigation in more complex mazes. While our main results focus on priors trained on weakly-informative data, we note that the performance of state-free priors with respect to behavioral priors is dependent on the choice of training task. We further elaborate on this topic by providing an additional empirical analysis on a more complex, object-interaction training task in Appendix D.

On the other hand, `gym-mujoco` environments do not offer a straightforward way to define tasks, and are therefore used to evaluate a proof-of-concept experiment on biased observations (Section 5.5).

More details can be found in Appendix J, while visual examples of environments are provided in Figure 3.

**Metrics and Training Data**   Our main metric for Sections 5.1, 5.2, 5.4, 5.5 is cumulative returns per test episode. For each task, we average this metric over 10 random seeds (excluding `gym-mujoco` experiments and visual RL experiments, which are respectively limited to 5 and 2 seeds for computational reasons). Unless stated explicitly, when aggregating results for different tasks (e.g. in plots marked as `downstream`), we perform a simple mean aggregation due to its statistical efficiency, as per-task scores are not normalized, and in the same range for all tasks. Alternatively, we report results for a more robust aggregation scheme (i.e. IQM (Agarwal et al., 2021)) in Appendix M. Uncertainty is quantified through 95% stratified basic bootstrap confidence intervals, as suggested in Agarwal et al. (2021). Across the experimental section, all priors are trained on 4000 trajectories of 500 steps each. An ablation on the amount of training data is reported in Appendix C. Further details can be found in Appendix K.1.

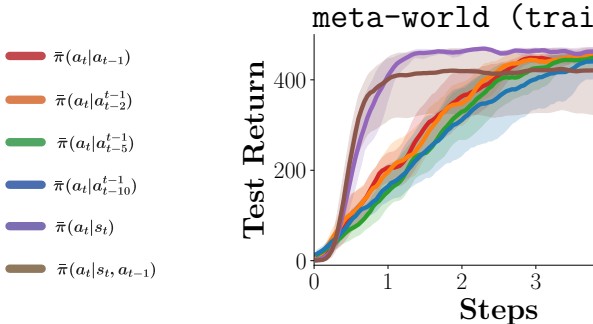
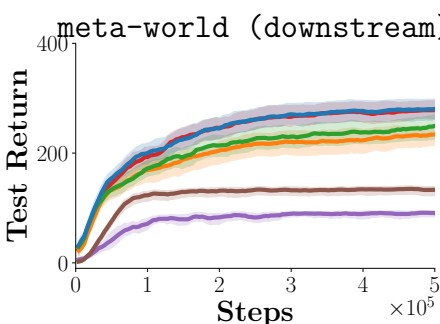

**Figure 4:** Comparison of downstream performance of action priors with different conditioning variables, reported on the training task (left) and averaged over downstream tasks (right).

## 5.1 Conditioning Variables for Action Priors

We now empirically show how the effectiveness of an action prior depends on the conditioning variables and on the similarity between the training and the downstream tasks. For this purpose, we train several variants of flow-based action priors on offline reaching trajectories and integrate them into a SAC learner as described in Section 4. In particular, we compare state-free priors conditioned on action sequences of different lengths (1, 2, 5, 10), non-Markovian priors conditioned on the previous state-action pair and a behavioral prior (conditioned on the state alone). In Figure 4 we report learning curves for the training task (`reach`), as well as averaged learning curves over all downstream tasks in `meta-world`. We observe that all priors are capable of guiding downstream RL as long as the task at hand matches the training task. In this case, state-conditional priors achieve slightly faster exploration by leveraging state-related and task-specific information. However, we find that including the state in the conditioning variables (as done in $\bar{\pi}(a_t|s_t)$ and $\bar{\pi}(a_t|s_t, a_{t-1})$) can jeopardize the ability to transfer knowledge to a different task. We hypothesize that this is due to a mismatch between the state distribution used for training the prior and that generated by the downstream task.

On the other hand, state-free priors are not conditioned on states by design, prove to be a capable alternative across both settings and are able to transfer knowledge to unseen tasks. While conditioning on longer action sequences can improve performance, we note that single-action-conditional models $\bar{\pi}(a_t|a_{t-1})$ are sufficient for capturing non-trivial temporal dependencies within our settings. Hence, they will be the focus of the remaining experiments.

## 5.2 Integration for State-free Priors

After showing a motivating application for state-free priors, we now argue the importance of our proposed mixture-based integration scheme for priors in offline reinforcement learning algorithms. Two common existing integration schemes for behavioral priors rely on regularizing the policy with a KL-term (Tirumala et al., 2020; Pertsch et al., 2020), or on a flow-based transformation of the action space (Singh et al., 2021). Both methods strongly build on the assumption that the prior is only conditioned on the current state $s_t$, or a subset thereof. In the case of flow-based action space warping, the prior is effectively integrated in the environment dynamics: as a result, if the prior is conditioned on previous actions $a_0^{t-1}$ or states $s_0^{t-1}$, the environment loses its Markov property, and stationary policies are no longer guaranteed to be optimal. On the other hand, penalizing the KL-divergence between the policy and a non-Markovian prior encourages the stationary policy to

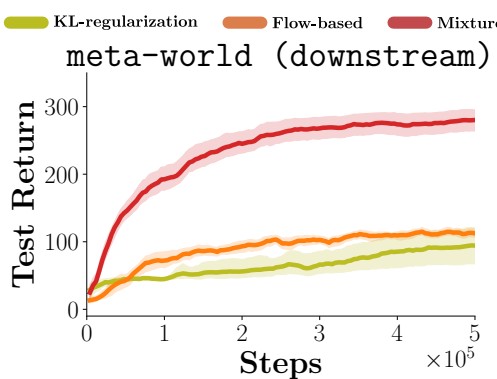

**Figure 5:** Performance of different integration schemes: our mixture-based integration compares favorably against existing methods (flow-based and KL-regularization).

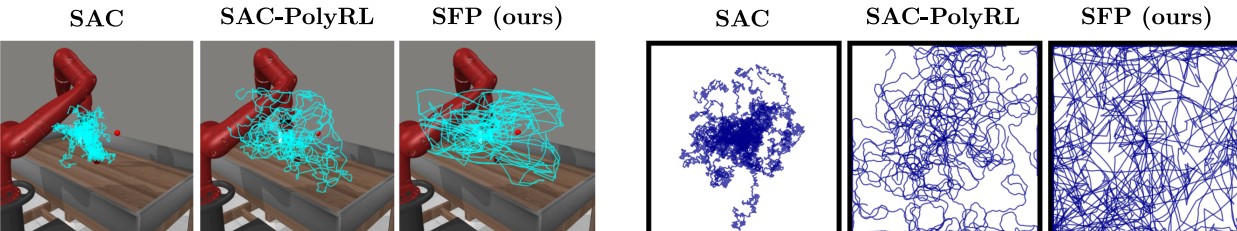

**Figure 6:** A qualitative comparison of sampled exploration trajectories in a robotic reaching task and in an empty 2D maze. Our method achieves directed behavior while covering most of the state space. SAC fails to cover the full state space, while SAC-PolyRL fails to reach distant areas consistently.

match the distribution of a potentially non-stationary prior, which is in general an ill-posed task. Instead, our mixture-based integration only suffers from biased learning objectives in the case of a non-Markovian action prior (see Appendix A). Empirically, we found this to be a mild limitation, as our integration achieves sensibly better performance compared to the two baselines in downstream learning, as reported in Figure 5, where experimental settings are the same as those described in Section 5.1, and details can be found in Appendix K.3.

## 5.3 Correlation and State Coverage

In this experiment, we show how a one-step state-independent state-free prior $\bar{\pi}(a_t|a_{t-1})$ can generate correlated and directed behavior, which leads to a more complete coverage of the state space during exploration. To this end, we sample 20 random trajectories of 500 steps each with our method and two relevant baselines in the `room` and `reach` environments.

As shown in Figure 6, the state-free prior produces directed behavior which covers most of the state space. As expected, uniform sampling (which approximates SAC's exploration) fails to reach the boundaries of the environment; SAC-PolyRL is capable of producing correlated and directed behaviors, but only after careful design and tuning. This qualitative assessment is consistent with the quantitative evaluation in Table 1, which reports state space coverage and radius of gyration squared (Amin et al., 2021) (see Appendix K.1 for details).

## 5.4 Transfer Learning

Our main results are obtained by comparing our method against several baselines in downstream learning tasks with a vectorial state space, as presented in Figure 7. We report learning curves averaged across downstream tasks, as well as on training tasks for reference. Results for each task are disentangled in Appendix L.

**Table 1:** State coverage metrics for our method and baselines. SFP's trajectories are locally directed and cover the state space well in both environments.

|      |            | $U_g^2$ | % Coverage |
|------|------------|---------|------------|
| reach | SAC | 0.006±0.001 | 0.137±0.01 |
|       | SAC-PolyRL | 0.025±0.001 | 0.272±0.01 |
|       | SFP (ours) | **0.053±0.009** | **0.357±0.02** |
| room | SAC | 0.005±0.001 | 0.333±0.02 |
|       | SAC-PolyRL | 0.026±0.002 | 0.880±0.04 |
|       | SFP (ours) | **0.054±0.008** | **0.963±0.04** |

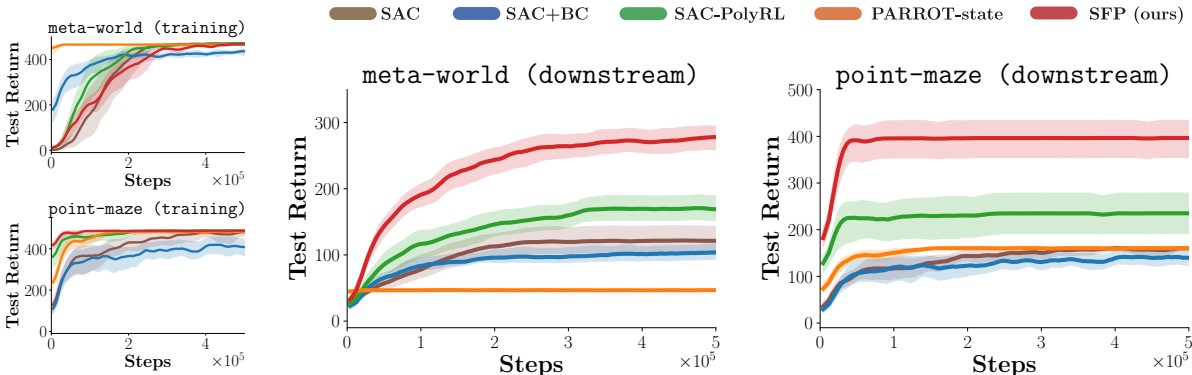

**Figure 7:** Accelerating downstream RL. While other methods are mostly competitive on training tasks (`reach` and `room`), SFP is better suited for accelerating RL in unseen tasks. Results are averaged over 10 seeds per task.

As expected, we observe that the performance of behavioral priors depends on the similarity of between the task at hand and expert demonstrations. This is the case for PARROT-state, which instantly solves the training tasks (`reach` and `room`), as its behavioral prior already represents a strong policy. On tasks which are significantly different from the training tasks, PARROT-state is however unable to guide the policy, as it receives out-of-distribution states, resulting in low average performance. On the other hand, SFP is largely more capable of transferring knowledge to unseen tasks, while rapidly catching up with PARROT-state in the training tasks.

Other baselines are in general less effective across the benchmarks: Vanilla SAC only makes progress on easier tasks, which can be achieved even with weak exploration.

SAC-PolyRL is able to produce good explorative trajectories through its hand-crafted policy and improves on SAC, but fails to explore after its initial phase and does not solve harder tasks. As previously reported by Singh et al. (2021), initializing SAC through behavioral cloning (SAC+BC) can help guide exploration without constraining the policy, but it fails to generalize across tasks and is regularly outperformed by stronger methods.

## 5.5 Additional Applications

We finally provide proofs of concept for possible applications of SFP. Per-task results are available in Appendix L.

**Visual RL** State-free priors allow the state space of the downstream task to be defined arbitrarily. Hence, they also allow transfer to the visual RL setting, which avoids reliance on a low-dimensional vectorized state space and is purely based on RGB observations. To this end, we compare SFP with Vanilla SAC and with PARROT in its original, visual setup, i.e., by conditioning its behavioral prior on images. We remark that,

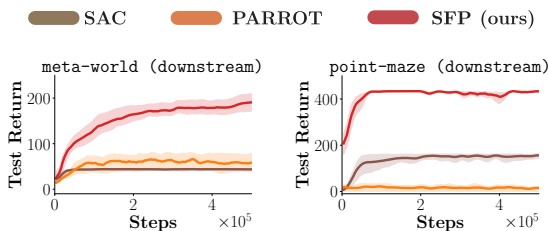

**Figure 8:** Performance on downstream learning in visual settings, averaged per suite.

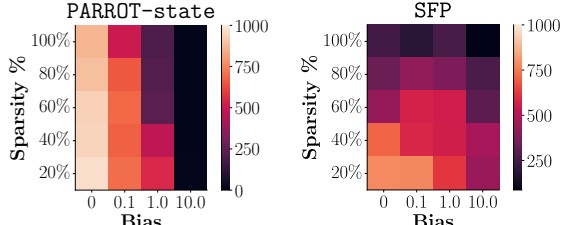

**Figure 9:** Mean performance on `gym-mujoco` environments for different sparsity thresholds and bias norms.

in this case, PARROT needs access to additional data (i.e., RGB observations) compared to SFP. We report the results in Figure 8. In general, we observe a performance drop for all methods compared to non-visual settings, which we hypothesize is due to the inherent challenge of learning in high-dimensional state spaces (Laskin et al., 2020). Nonetheless, our findings remain generally valid in visual settings, as SFP retains most of its exploration capability and, despite failing to make progress on a majority of visual tasks, consistently outperforms the baselines. Per-task results are reported in Appendix L.

**Biased Downstream Learning**   A final potential application for state-free priors is that of guiding exploration when downstream learning is corrupted by biased observations. This scenario could model a form of sim-to-real transfer, in which training trajectories were cheaply collected in simulation, but the physical testbed was miscalibrated, resulting in biased observations. We provide experimental results in this setup based on `gym-mujoco` environments (`Ant-v2`, `HalfCheetah-v2`, `Swimmer-v2`, `Hopper-v2`, `Walker2d-v2`). Training trajectories are collected by a SAC agent that was fully trained on dense rewards. In downstream learning, the reward is instead sparse (i.e. only positive when the displacement of the CoM from its initial position surpasses a sparsity threshold $\bar{x}$), and significant exploration is required. Moreover, the observations in downstream learning are corrupted by a fixed additive bias $s = s + b$. Figure 9 compares test performance of downstream agents when guided by a behavioral prior (PARROT-state) or by a state-free prior (SFP), after 500k steps and averaged across environments. We observe that behavioral priors are capable of extracting a near-optimal policy from the training data, which transfers perfectly to the sparsified environments if the observations are uncorrupted. However, as bias increases, their performance sharply drops. On the other hand, state-free priors are less effective in unbiased conditions, but their performance is generally retained as bias increases. More details on this experiment are presented in Appendix J.3, and an ablation introducing a transformation on the action space is reported in Appendix E.

## 6   Discussion and Conclusion

**Limitations**   While state-free priors are able to generalize to a broader set of tasks with respect to behavioral priors, they are data-driven, and as such will only reconstruct behavior that appears in the offline dataset of trajectories they are trained on. For instance, when training on demonstrations for `reach`, state-free priors will not encode any grasping strategy. While such behavior can still be recovered by the policy $\pi$, there is no incentive to do so. As a result, successfully transferring to tasks that require additional strategies, such as grasping (e.g., `pick-place`) remains an open challenge. Secondly, as shown in Figure 7, behavioral priors remain superior when the downstream state distribution is not distant from the state distribution observed in the training task. As state-free priors do not model state-action relationships, they can only provide weaker guidance in this privileged case. This is in particular true when priors are trained on expert trajectories for complex manipulation tasks, as shown in Appendix D. For this reason, a strategy to combine benefits from behavioral and state-free priors would represent an interesting direction for the future.

**Conclusion**   In this paper we proposed a method for improving exploration efficiency in off-policy reinforcement learning. In particular, we introduced state-free priors, a family of non-Markovian, state-independent and flow-based action priors, and proposed a principled manner for integration into an off-policy reinforcement learning framework. While leveraging a relatively simple idea, state-free priors represent a powerful method for exploration. As shown by significant empirical evidence, they can massively accelerate learning across a diverse set of tasks, while solely requiring a modest amount of offline, weakly informative trajectories.

**Broader Impact Statement**

Our main contribution revolves on accelerating and enabling reinforcement learning in environments with a strong exploration component. As a consequence, we believe that concerns with respect to our method are for the most part aligned with general RL research. For instance, improved sample efficiency could on one hand accelerate the process of automation, which might have a negative impact on societal equality, and on the other hand democratize access to powerful RL methods by lowering the amount of required resources. Due to the general nature of our method, we believe that we do not introduce fundamentally new risks.

**Acknowledgments**

We are grateful to Núria Armengol Urpí, Emre Aksan and others for providing constructive feedback on this work. We also acknowledge the anonymous reviewers for their important contributions towards improving the manuscript.

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

## A  Objective Derivation

The purpose of this section is that of describing how learning objectives for the policy $\pi$, the Q-estimator $Q^{\tilde{\pi}}$ and the mixing function $\Lambda$ can be derived from a max-entropy objective; while in practice each function is respectively parameterized by $\phi, \theta$ and $\omega$, in order to ease notation we will only explicitly mention these parameters when introducing empirical loss estimates. The objective is defined as

$$\mathbb{E}_{\tau \sim \tilde{\pi}}\left[\sum_{t=0}^{\infty}\left(\gamma^t R(s_t, a_t) + \alpha \mathcal{H}(\pi(\cdot|s_t))\right)\right], \tag{8}$$

where the expectation is computed over trajectories generated by the mixture $\tilde{\pi} = (1-\lambda_t)\pi + \lambda_t\bar{\pi}$, $\bar{\pi}$ is a fixed action prior and $\lambda_t = \Lambda(s_t)$ is the mixing weight. The optimization of Equation 8 can be performed by modeling the action prior $\bar{\pi}(\cdot)$ as a Markovian distribution over the action space $\mathcal{A}$; we note that non-Markovian action priors do not necessarily satisfy Equation 10 and therefore can introduce bias in Q-value targets (Equation 12). The steps of these derivations follow those reported in Haarnoja et al. (2018), as our method is designed to fit within the proposed framework. First, let us formally introduce the Q-function for the mixture $\tilde{\pi}$:

$$Q^{\tilde{\pi}}(s, a) = \mathbb{E}_{\tau \sim \tilde{\pi}}\left[\sum_{t=0}^{\infty}\left(\gamma^t R(s_t, a_t)\right) + \alpha \sum_{t=1}^{\infty}\left(\gamma^t \mathcal{H}(\pi(\cdot|s_t))\right)|s_0 = s, a_0 = a\right]. \tag{9}$$

We can then formulate the Bellman Equation and explicitly unravel the entropy term:

$$\begin{aligned}
Q^{\tilde{\pi}}(s, a) &= \mathbb{E}_{\substack{s' \sim \mathcal{T}(\cdot|s,a) \\ \tilde{a}' \sim \tilde{\pi}(\cdot|s')}}\left[R(s, a) + \gamma\left(Q^{\tilde{\pi}}(s', \tilde{a}') + \alpha \mathcal{H}(\pi(\cdot|s'))\right)\right] \\
&= \mathbb{E}_{\substack{s' \sim \mathcal{T}(\cdot|s,a) \\ \tilde{a}' \sim \tilde{\pi}(\cdot|s')}}\left[R(s, a) + \gamma\left(Q^{\tilde{\pi}}(s', \tilde{a}') + \alpha \mathbb{E}_{a' \sim \pi(\cdot|s')}[-\log(\pi(a'|s'))]\right)\right] \\
&= \mathbb{E}_{\substack{s' \sim \mathcal{T}(\cdot|s,a) \\ \tilde{a}' \sim \tilde{\pi}(\cdot|s') \\ a' \sim \pi(\cdot|s')}}\left[R(s, a) + \gamma\left(Q^{\tilde{\pi}}(s', \tilde{a}') - \alpha \log(\pi(a'|s'))\right)\right].
\end{aligned} \tag{10}$$

The right-hand side of Equation 10 can be estimated via Monte Carlo sampling and used as a target for Q-estimates, which can then be trained by minimizing a standard MSE loss. In practice, as is done for SAC, two separate parameterized Q-function estimators $(Q^{\tilde{\pi}}_{\theta_i})_{i=1,2}$ are used to prevent overestimating Q-values. Additionally, we also adopt target networks $(Q^{\tilde{\pi}}_{\theta_{\text{target},i}})_{i=1,2}$, updated via Polyak averaging. As a result, when sampling a batch $B$ from a replay buffer, the empirical estimates for Q-losses are:

$$\hat{J}_{Q_{\theta_i}} = \frac{1}{|B|} \sum_{(s,a,r,s',d) \in B}\left(Q^{\tilde{\pi}}_{\theta_i}(s, a) - \hat{y}_t(s', r, d)\right)^2, \tag{11}$$

where the target for the Q-value is computed as

$$\hat{y}_t(s', r, d) = r + \gamma(1 - d)\left(\min_{i=1,2} Q^{\tilde{\pi}}_{\theta_{\text{target},i}}(s', \tilde{a}') - \alpha \log \pi_\phi(a'|s')\right) \quad \text{with } \tilde{a}' \sim \tilde{\pi}(\cdot|s'), a' \sim \pi_\phi(\cdot|s'). \tag{12}$$

and $r, d$ stand for the reward and done signal, respectively.

As the prior $\bar{\pi}$ is fixed, optimizing the mixture policy $\tilde{\pi} = \lambda\bar{\pi} + (1-\lambda)\pi$ only involves the optimization of $\pi$ and $\lambda$. The policy $\pi$ can be trained to minimize the KL-divergence with the soft-max of the Q-function, or alternatively to maximize the value function (Haarnoja et al., 2018). Our method relies on the second option and trains both $\pi$ and $\Lambda$ to maximize the value function $V^{\tilde{\pi}}(s)$, which can be formulated as follows:

$$V^{\tilde{\pi}}(s) = \mathbb{E}_{\substack{\tilde{a} \sim \tilde{\pi}(\cdot|s) \\ a \sim \pi(\cdot|s)}}\left[Q^{\tilde{\pi}}(s, \tilde{a}) - \alpha \log(\pi(a|s))\right]. \tag{13}$$

Interestingly, the value function $V^{\tilde{\pi}}$ can be decomposed as

$$
\begin{aligned}
V^{\tilde{\pi}}(s) &= \underset{\tilde{a}\sim\tilde{\pi}(\cdot|s)}{\mathbb{E}}\left[Q^{\tilde{\pi}}(s,\tilde{a})\right] - \alpha \underset{a\sim\pi(\cdot|s)}{\mathbb{E}}\left[\log(\pi(a|s))\right] \\
&= \left(\int_{\tilde{a}\in\mathcal{A}} Q^{\tilde{\pi}}(s,\tilde{a})\tilde{\pi}(\tilde{a}|s)\right) - \alpha \underset{a\sim\pi(\cdot|s)}{\mathbb{E}}\left[\log(\pi(a|s))\right] \\
&= \left(\int_{\tilde{a}\in\mathcal{A}} Q^{\tilde{\pi}}(s,\tilde{a})\big(\lambda\bar{\pi}(\tilde{a}) + (1-\lambda)\pi(\tilde{a}|s)\big)\right) - \alpha \underset{a\sim\pi(\cdot|s)}{\mathbb{E}}\left[\log(\pi(a|s))\right] \\
&= \lambda\left(\int_{\tilde{a}\in\mathcal{A}} Q^{\tilde{\pi}}(s,\bar{a})\bar{\pi}(\bar{a})\right) + (1-\lambda)\left(\int_{a\in\mathcal{A}} Q^{\tilde{\pi}}(s,a)\pi(a|s)\right) - \alpha \underset{a\sim\pi(\cdot|s)}{\mathbb{E}}\left[\log(\pi(a|s))\right] \\
&= \lambda \underset{\bar{a}\sim\bar{\pi}(\cdot)}{\mathbb{E}}\left[Q^{\tilde{\pi}}(s,\bar{a})\right] + (1-\lambda) \underset{a\sim\pi(\cdot|s)}{\mathbb{E}}\left[Q^{\tilde{\pi}}(s,a)\right] - \alpha \underset{a\sim\pi(\cdot|s)}{\mathbb{E}}\left[\log(\pi(a|s))\right],
\end{aligned}
\tag{14}
$$

where $\lambda = \Lambda(s)$ is the mixing weight computed on the current state. When differentiating this objective with respect to the parameters $\phi$ of the policy network $\pi$, the first term does not contribute to gradients, resulting in

$$
\max_{\phi} V^{\tilde{\pi}}(s) = \max_{\phi} \underset{a\sim\pi_\phi(\cdot|s)}{\mathbb{E}}\left[(1-\lambda)Q_\theta^{\tilde{\pi}}(s,a) - \alpha\log(\pi_\phi(a|s))\right].
\tag{15}
$$

Computing the expectation over actions can then be circumvented by using the reparameterization trick, which enables expressing actions as $a_\phi(\xi,s)$, where $\xi\sim\mathcal{N}$ is sampled from a standard Gaussian. In practice, considering the two Q-networks, the policy network can be optimized by minimizing the empirical loss estimated over a batch $B$:

$$
\hat{J}_{\pi_\phi} = \frac{1}{|B|}\sum_{(s)\in B}\left[(1-\Lambda_\omega(s))\big(\min_{i=1,2} Q_{\theta_i}^{\tilde{\pi}}(s,\pi_\phi(s)) - \alpha\log(\pi_\phi(a|s))\big)\right] \quad \text{with } a\sim\pi_\phi(\cdot|s),
\tag{16}
$$

which can be minimized via standard first-order optimization techniques.

The last learning objective to recover is that for the parameterized mixing function $\Lambda_\omega$. Once again, we obtain this by maximizing the value function $V^{\tilde{\pi}}(s)$. Starting from Equation 14, we can now drop the final term, which does not depend on $\omega$:

$$
\begin{aligned}
\max_{\omega} V^{\tilde{\pi}}(s) &= \max_{\omega} \Lambda_\omega(s) \underset{\bar{a}\sim\bar{\pi}(\cdot)}{\mathbb{E}}\left[Q_\theta^{\tilde{\pi}}(s,\bar{a})\right] + (1-\Lambda_\omega(s)) \underset{a\sim\pi_\phi(\cdot|s)}{\mathbb{E}}\left[Q_\theta^{\tilde{\pi}}(s,a)\right] \\
&= \max_{\omega} \Lambda_\omega(s) \underset{\bar{a}\sim\bar{\pi}(\cdot)}{\mathbb{E}}\left[Q_\theta^{\tilde{\pi}}(s,\bar{a})\right] + \underset{a\sim\pi_\phi(\cdot|s)}{\mathbb{E}}\left[Q_\theta^{\tilde{\pi}}(s,a)\right] - \Lambda_\omega(s) \underset{a\sim\pi_\phi(\cdot|s)}{\mathbb{E}}\left[Q_\theta^{\tilde{\pi}}(s,a)\right] \\
&= \max_{\omega} \Lambda_\omega(s) \underset{\bar{a}\sim\bar{\pi}(\cdot)}{\mathbb{E}}\left[Q_\theta^{\tilde{\pi}}(s,\bar{a})\right] - \Lambda_\omega(s) \underset{a\sim\pi_\phi(\cdot|s)}{\mathbb{E}}\left[Q_\theta^{\tilde{\pi}}(s,a)\right] \\
&= \max_{\omega} \Lambda_\omega(s) \underset{\substack{\bar{a}\sim\bar{\pi}(\cdot|s)\\a\sim\pi_\phi(\cdot)}}{\mathbb{E}}\left[Q_\theta^{\tilde{\pi}}(s,\bar{a}) - Q_\theta^{\tilde{\pi}}(s,a)\right].
\end{aligned}
\tag{17}
$$

In practice, the loss estimate for training the mixing network $\Lambda_\omega$ can be computed over a batch $B$ as

$$
\hat{J}_{\Lambda_\omega} = \frac{1}{|B|}\sum_{(s)\in B}\left[\Lambda_\omega(s)\big(\min_{i=1,2} Q_{\theta_i}^{\tilde{\pi}}(s,\bar{a}) - \min_{i=1,2} Q_{\theta_i}^{\tilde{\pi}}(s,a)\big)\right] \quad \text{with } \bar{a}\sim\bar{\pi}(\cdot), a\sim\pi_\phi(\cdot|s).
\tag{18}
$$

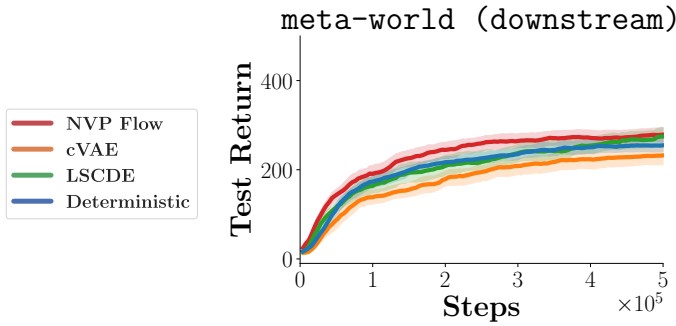

**Figure 10:** Performance on downstream `meta-world` tasks when modeling a one-step state-free prior through various generative models.

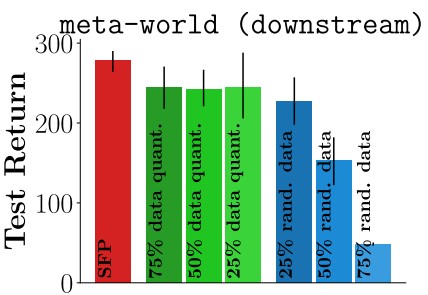

**Figure 11:** Performance on downstream `meta-world` tasks when training on fewer offline demonstrations, or when they are corrupted by uniform noise.

The gradient of this objective with respect to the mixing weight $\lambda = \Lambda(s)$ is intuitively the difference in Q-values between actions drawn from the prior and policy. As training proceeds, and the policy $\pi$ improves, the gradient turns negative and encourages lower mixing weights, gradually abandoning the prior $\bar{\pi}$'s guidance.

## B  Model Ablation

We now set out to provide empirical backing for an important design choice. Our policy mixing approach grants freedom in choosing generative models capable of describing complex distributions, as computing distance metrics to the prior is not required. We compare the performance of Real NVP Flows with a Conditional VAE (Sohn et al., 2015), a non-parametric conditional Least Squares Density Estimator (Sugiyama et al., 2010), and an MLP modeling a deterministic prior. As shown in Figure 10, we found flow-based modeling to be both competitive and practical.

## C  Data Ablation

Figure 11 includes data quantity ablations (with $75\%, 50\%$ and $25\%$ of training data), and data quality ablations (by substituting $25\%, 50\%$ and $75\%$ of the training data with uniformly sampled actions). In the latter case, we observe a sharp drop in final performance (averaged over downstream `meta-world` tasks), while we find SFP to remain viable in low-data regimes.

## D  Training Task Ablation

The effectiveness of a learned prior depends on the nature of both training, and downstream tasks. In the context of this paper, we focus on simple training tasks, which can be solved easily by existing RL algorithms (e.g. `reach` from `meta-world`). In this section, however, we instead evaluate several prior designs when training on expert trajectories from a more complex and informative task (i.e. `pick-place`). We set out to verify whether (a) state-free priors can fail to exploit high-quality expert training trajectories due to their lack of expressiveness, and (b) how state-free priors compare to behavioral priors on unseen tasks whose state distribution still matches the expert state distribution (i.e. `push`).

Our experimental comparison includes SFP ($\bar{\pi}(a_t|a_{t-1})$), PARROT-state, as well as two variants of SFP that involve priors conditioned on the current state ($\bar{\pi}(a_t|s_t)$), or on the current state and previous action ($\bar{\pi}(a_t|s_t, a_{t-1})$). Figure 12 gathers the performance of all methods in the training task (`pick-place`, left), averaged over all downstream `meta-world` tasks (middle), and on a downstream task which is unseen, but remains in-distribution of the training task (`push`, right).

We observe that only state-conditioned approaches are able to make progress on the training task, as $\bar{\pi}(a_t|a_{t-1})$ lacks sufficient expressiveness for guiding exploration, since its state-free prior cannot encode

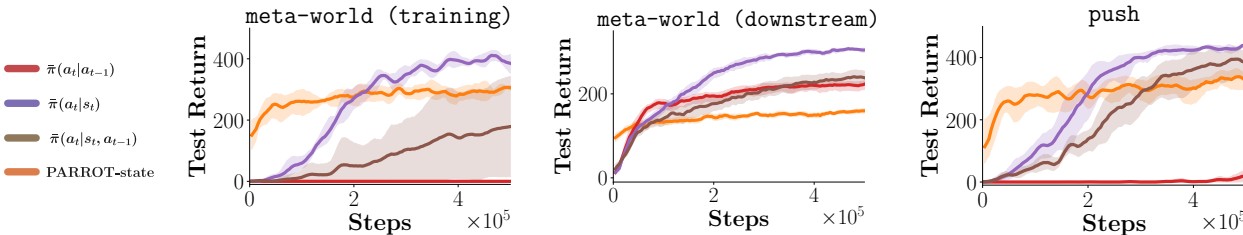

**Figure 12:** When trained on a task involving manipulation, state-conditioned priors can succeed in complex tasks (the training task `pick-place` on the left, and an unseen in-distribution task `push` on the right), while $\bar{\pi}(a_t|a_{t-1})$ fails. However, when averaging returns over all downstream tasks (middle), $\bar{\pi}(a_t|a_{t-1})$ can partially close the gap.

closed-loop interaction with objects. Similarly, when evaluating on an unseen in-distribution task requiring manipulation (`push`), we observe that behavioral priors are, as expected, able to learn the task, while $\bar{\pi}(a_t|a_{t-1})$ fails to find a solution. However, when performance is averaged over all downstream tasks, we find that the robustness to domain shifts compensates the lack of expressiveness, and $\bar{\pi}(a_t|a_{t-1})$'s performance is roughly comparable with other methods.

Within state-conditional approaches, we note that the two methods adopting our integration scheme can dynamically manage the frequency to which the prior is sampled, and can ignore the prior when it suggests suboptimal solutions. As a consequence, their performance on downstream tasks is, on average, improved. PARROT-state, on the other hand, relies on a hard integration of the prior, and is the most sample efficient method for in-distribution tasks, but suffers largely when dealing with out-of-distribution downstream tasks.

Finally, we observe that priors conditioned on action-state pairs $\bar{\pi}(a_t|s_t, a_{t-1})$ are underperforming with respect to purely behavioral priors $\bar{\pi}(a_t|s_t)$ in this setting. Considering the results reported in Section 5.1, which show an inverted pattern, we note that conditioning the prior on action-state pairs allows it to extract conditional dependencies on both the state space and the action space. Such priors are not robust to distribution shifts in the state space, but may also learn to focus more on their action inputs. As a consequence, we hypothesize that their average performance generally falls between that of purely state-free, and purely behavioral priors.

# E Ablation on Biased Downstream Learning

The experiment presented in Section 5.5 introduces a potential application of state-free priors in case the downstream task's state space is corrupted by bias. We now study the setting in which the state space of training and downstream tasks matches, but the action space is disrupted through a transformation. For simplicity, we consider a permutation of the action vector, and in particular an inversion (i.e. `action=action[::-1]`), which occurs before the action is executed in the environment. This transformation

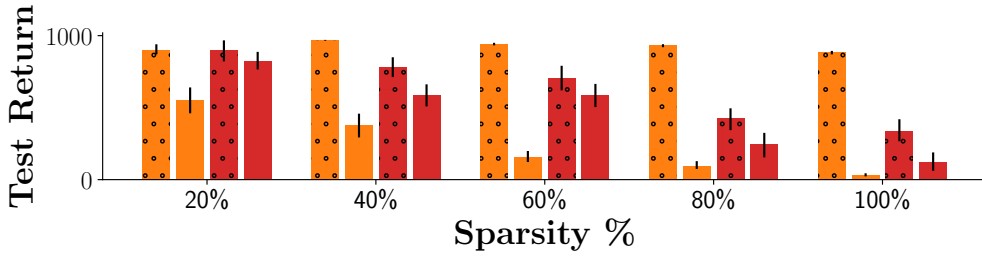

**Figure 13:** SFP (red) and PARROT-state (orange) performance aggregated over mujoco tasks. Dotted bars represent performance on the unmodified training task, and undotted bars represent performance on the training task, when introducing a disruptive transformation of the action space.

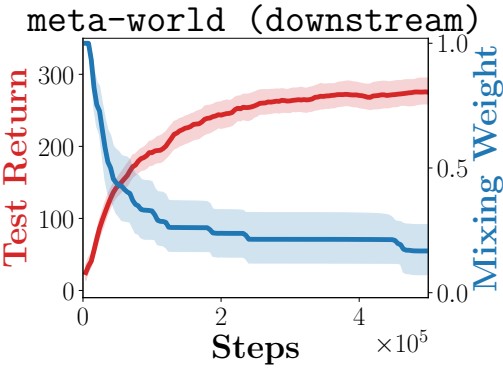
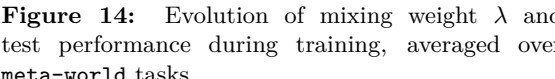
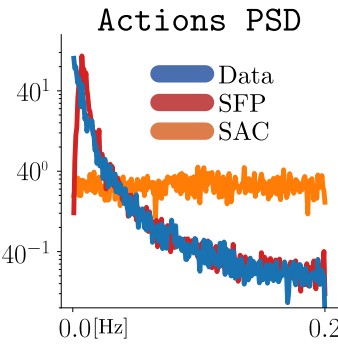

**Figure 14:** Evolution of mixing weight $\lambda$ and test performance during training, averaged over `meta-world` tasks.

**Figure 15:** PSD of trajectories sampled from the offline dataset, from SFP's prior and from a uniform random actor.

is chosen because, contrary to adding noise to the state space, it would not affect the performance of a naive RL algorithm (e.g. SAC), while potentially impacting the effectiveness of learned priors.

In Figure 13 we report final returns for SFP and PARROT-state averaged over the five `mujoco` environments used in Section 5.5. We evaluate different degrees of reward sparsity (as explained in Section 5.5), and for each one we compare the performance of SFP and PARROT-state on the training task, as well as on a corrupted version of the training task. We note that the added transformation results in a comparable drop in performance for both methods. In the case of SFP, this can be partially traced back to the fact that the prior may be conditioned on out-of-distribution actions. Most importantly, for both SFP and PARROT-state, this transformation disrupts the actions suggested by the prior, as the trajectories it was trained on cannot be reproduced in the environment.

## F   Study on Learned Mixing Weights

In Figure 14, we plot the mixing weight $\lambda$ during the course of training against returns, averaged over the downstream `meta-world` tasks. As formally described in Section 4, the mixing weight generally decreases as returns increase; in particular, we observe $\lambda \to 0$ for solved tasks and $\lambda \to 1$ for unsolved tasks. We found this trend to be consistent for most states visited. Per-task plots are reported in Appendix L. We also note that the initial trajectory is influenced by the initialization discussed in Appendix K.4.

## G   Study on Temporal Correlation

In order to better quantify the concept of temporally correlated exploration, we plot the PSD (averaged over action dimensions) of action sequences sampled from the training set, from our learned prior, as well as those obtained by sampling from a uniform distribution on `room`. In Figure 15 we note prominent low frequencies for both the training data and SFP trajectories.

## H   Additional Baselines

In this section we report results for an additional baseline, namely SPIRL (Pertsch et al., 2020). SPIRL is a hierarchical method that combines skill learning with a high-level behavioral prior, which is integrated in a SAC agent via KL-regularization. We evaluate this method on the `meta-world` suite in the same settings reported in Section 5.4, namely by training all priors on reaching trajectories for random goals, and then reporting performance on the training task, as well as averaged over all downstream tasks. We also report the performance of SFP and PARROT-state from Section 5.4 as a reference.

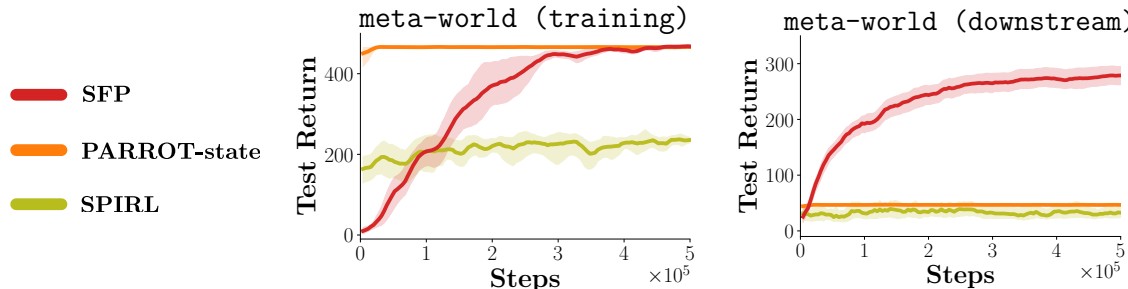

**Figure 16:** Comparison of SPIRL with SFP, as well as with another method based on behavioral priors (PARROT-state), reported on the training task (left) and averaged over downstream tasks (right). The setting is the same as for Figure 7.

Figure 16 highlights that, similarly to PARROT-state, SPIRL can instantly solve the training task, but suffers when transferred to unseen tasks with out-of-distribution states. We note that the lower performance in the trained task can be traced back to the hierarchical nature of SPIRL: its low level skill decoder is trained on reaching trajectories, which do not include static behavior. As a consequence, the agent has no access to a skill that keeps the gripper still, while `reach` requires the agent to hold the gripper in its goal position for several steps. Finally, we note that, while this skill-based approach needs further assumption on the training data to ensure learning all necessary skills, the soft integration of the behavioral prior could in principle allow the agent to mitigate issues related to out-of-distribution states.

This experiments relies on the official implementation (Pertsch et al., 2020), and adopts the hyperparameter set provided for Frankakitchen experiments. Further tuning of the target divergence parameter did not lead to significant improvements in performance.

# I    Additional Related Works

**Meta-RL and Task Inference**    The idea of quickly adapting to unseen tasks is a fundamental concept in meta-reinforcement learning (Schmidhuber, 1987; Wang et al., 2017). While SFP has a similar goal, it crucially relies on training a policy from scratch for downstream learning and is not designed for zero-shot adaptation. A line of research in meta-RL, referred to as *context-based* (Wang et al., 2017; Mishra et al., 2018; Rakelly et al., 2019), performs task inference in order to identify the current task and quickly extrapolate how to maximize returns. A task representation can in practice be extracted from recent experiences. Interestingly, state-free priors can in principle handle the same type of input, i.e. sequences of recent action-state pairs. However, instead of producing an explicit representation of the current task, state-free priors directly model an action distribution.

**Learning from Demonstrations**    SFP is aligned with existing methods that rely on demonstrations to accelerate RL on complex tasks (Nair et al., 2018; Rajeswaran et al., 2018; Christen et al., 2019). In principle, the cost of acquiring few expert trajectories can be small compared to the significant engineering effort required in their absence (OpenAI et al., 2019). In most cases, such trajectories are required to be near-optimal and collected in the same environment (Schaal, 1997) or from the same distribution of tasks (Singh et al., 2021). In our case, expert trajectories can be weakly informative , unlabeled and collected on a significantly different task, as we focus on reconstructing correlated exploration trajectories instead of exploitative behavior.

## J Environment Details

### J.1 Robot Manipulation

We rely on the `meta-world` suite (Yu et al., 2020) for robot manipulation experiments. It consists of a simulated 7 DoF Sawyer arm, implemented in the MuJoCo physics engine (Todorov et al., 2012). By default, states are represented as 36-dimensional vectors across all tasks. The state contains the 3D location and aperture of the gripper, the 3D location and quaternion of one object (e.g. door or window), measured for the current and previous time step. Goals are represented as the desired 3D location of the end effector or object, according to the task. Actions are 4-dimensional vectors containing a 3D movement and a 1D control over the aperture of the effector.

We additionally render 64x64 RGB images as observations for the visual setup in Section 5.5, using the camera angle `corner` (as can be seen in Figure 3). Instead of the originally proposed dense reward, we adopt a binary sparse reward which is non-zero only upon task completion. For more details, we refer to the original implementation (Yu et al., 2020).

### J.2 Maze Navigation

Our `point-maze` environments are adapted from Pitis et al. (2020). The state consists of the 2D position of the agent, which can be actuated via a velocity-controller through 2D actions. The goal space matches the state spaces in dimensions and representation. The reward signal is 1 when the Euclidean distance to the goal is lesser than 1.2 units, else it is 0. We experiment with two different layouts (renderings can be found in Figure 3):

- `room` is a large, 29×29 square room. The starting position is at the center, and the goal is initialized randomly in one of the 4 corners at each reset. Trajectories for training the priors are obtained from this environment. Only for Figure 6, the size was increased to 81×81, to allow and better visualize long trajectories.

- `corridor` is a larger u-shaped corridor with three parts of lengths 60, 3 and 60 respectively, assembled at 90° clockwise rotations. The structure of the corridor can therefore be contained in a 60 × 3 rectangle. Starting and goal position are fixed and located at opposite ends of the corridor.

- `maze` is a maze presenting intersections and dead ends. Starting and goal positions are fixed, and the optimal policy requires 66 steps.

For further details, we refer directly to the published codebase (URL in Footnote 1).

### J.3 Continuous Control

Due to their prevalence as a benchmark, we use `gym-mujoco` environments to evaluate SFP and baselines in the presence of biased observations. We evaluate methods across 5 popular environments, namely `Ant-v2`, `HalfCheetah-v2`, `Hopper-v2`, `Swimmer-v2` and `Walker2d-v2` (see Figure 17). Additionally, we parameterize each environment with two variables, namely the bias $b$ and the sparsity parameter $\bar{x}$. Observations in downstream learning are corrupted as $s \leftarrow s + b$, and rewards are sparsified as $r_t = \mathbf{1}_{x \geq \bar{x}}$ where $x$ is the L2 distance between the initial and current position of the CoM. In order to aggregate data across environments, we empirically find an environment-specific maximum sparsity threshold $\bar{x}_{max}$ for which SFP fails to reach meaningful rewards. For each environment, we then evaluate all methods on a set of 6 sparsity coefficients scaled linearly from 0 to $\bar{x}_{max}$, that are referred to as percentages in Figure 9 and 23. The maximum sparsity thresholds $\bar{x}_{max}$ is 12 for `HalfCheetah-v2`, 2 for `Ant-v2`, `Hopper-v2` and `Swimmer-v2` and 1 for `Walker-v2`.

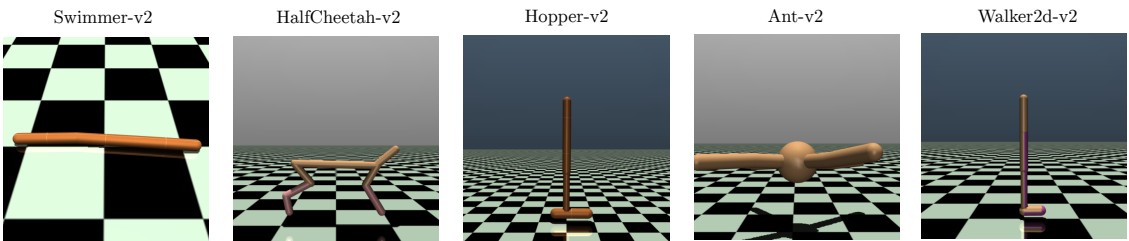

| Swimmer-v2 | HalfCheetah-v2 | Hopper-v2 | Ant-v2 | Walker2d-v2 |

**Figure 17:** Overview of the different `gym-mujoco` environments presented in Section 5.5.

## K   Implementation Details

### K.1   Metrics

All metrics reported in this paper (e.g. cumulative returns per episode) are averaged over 10 random seeds (excluding `gym-mujoco` experiments and visual RL experiments, which are respectively limited to 5 and 2 seeds for computational reasons); mean learning curves are smoothed over 5 steps and shaded areas represent standard deviation across seeds.

We rely on two metrics for measuring the quality of exploration trajectories in Section 5.3:

- **%Coverage** divides the reachable state space in $n$ cubic buckets ($n = 1000$ for `reach` and $n = 100$ for `room`) and reports the ratio between the number of buckets visited by a set of 20 trajectories of length 500 and the total number of buckets.

- **Radius of Gyration Squared** measures the spread in visited states, averaged over all trajectories, and is adapted from Amin et al. (2021). Given a set $T$ of $n$ trajectories, the metric can be computed as:

$$U_g^2(T) = \frac{1}{\delta n} \sum_{\tau \in T} \frac{1}{|\tau| - 1} \sum_{s \in \tau} d^2(s, \bar{\tau}),$$

  where a trajectory $\tau$ is modeled as a sequence of states $(s_i)_0^{|\tau|-1}$, $d^2(\cdot, \cdot)$ measures the Euclidean distance and $\bar{\tau} = \frac{1}{|\tau|} \sum_{s \in \tau} s$. We additionally normalize the metric by $\delta$, which measures the diagonal of the box containing reachable states.

### K.2   Data Collection

Training tasks for `point-maze` and `meta-world` are relatively simple, and allow collecting demonstrations through a scripted policy. Since these environments do not include obstacles and allow direct control over the agent position (in 2D for `room` and in 3D for `reach`), scripted policies simply receive the 2D/3D positions of the agent and of its goal, and output a distance vector. This vector is then corrupted with isotropic Gaussian noise and scaled to fit within action limits. A simple implementation of scripted policies is available as part of our codebase (see URL in Footnote 1). For each training environment (`reach` or `room`), we collect 4000 trajectories of 500 steps each. Goals for the scripted policy are sampled uniformly from the reachable state space. On goal achievement, a new goal is sampled, once again uniformly. In the context of `gym-mujoco` experiments, 1000 trajectories of 500 steps each are instead collected by fully-trained SAC Haarnoja et al. (2018) agents. The same expert datasets are then used for training state-free priors, behavioral priors or behavioral cloning.

### K.3   Baselines

**SAC**   We build upon the implementation provided by SpinningUp (Achiam, 2018), which is reported to be roughly on-par with the best results achieved on `gym-mujoco` (Brockman et al., 2016). For simplicity, we do not use automatic entropy tuning and keep $\alpha$ constant during learning. Nonetheless, our method could easily tune $\alpha$ dynamically at the expense of increased complexity, as is described in Haarnoja et al. (2018).

**Table 2:** Hyperparameters for SAC.

| Hyperparameter | Value |
|---|---|
| Epochs | 125 |
| Steps Per Epoch | $4e3$ |
| Steps of Initial Exploration | $1e4$ |
| Steps Before Training | $1e3$ |
| Environment Steps per Iteration | 50 |
| $\gamma$ | 0.99 |
| Polyak Averaging Rate | 0.995 |
| Inverse of Reward Scale ($\alpha$) | 0.2 for META-WORLD and GYM-MUJOCO, 0.02 for POINT-MAZE |
| Batch Size | 100 |
| Optimizer | Adam |
| $\beta_1$ | 0.9 |
| $\beta_2$ | 0.999 |
| Learning Rate | 0.001 |
| Hidden Units | 256 |
| Hidden Layers | 2 |
| Hindsight Replay Ratio | 4 |
| Replay Size | $5e5$ for vector-based RL, $2e5$ for image-based RL |

Moreover, we introduce n-step returns (Hessel et al., 2018) with $n = 10$. We experimented with importance sampling for off-policy correction, but, similarly to what is reported by Hessel et al. (2018), we observed no empirical benefit. All remaining hyperparameters are reported in Table 2. We anticipate that all baselines also rely on n-step returns.

**SAC+BC**  Behavioral cloning (BC) is performed on the entire dataset $\mathcal{D}$ for 10 epochs, by maximizing the log-likelihood of the Gaussian policy with respect to expert actions. The optimizer and batch size used for BC are the same as for downstream learning.

**SAC-PolyRL**  SAC-PolyRL (Amin et al., 2021) replaces the initial uniform exploration phase of SAC with trajectories collected by a hand-crafted policy. While SAC's hyperparameters are unvaried, SAC-PolyRL specific parameters are tuned from those reported in various settings in the original paper. We use $\theta = 0.35$, $\sigma^2 = 0.017$ and $\beta = 0.01$.

**PARROT**  An official implementation for PARROT (Singh et al., 2021) is not available at the time of writing. We reproduce the training routine and downstream application and adopt all hyperparameters reported in the original paper. Generative models are trained until convergence (100 epochs) using a batch size of 400 samples and Adam (Kingma & Ba, 2015) as an optimizer, with a learning rate of 0.0001, $\beta_1 = 0.9$, $\beta_2 = 0.999$ and a weight decay penalty of $1e - 6$.

In addition to the image-based version of PARROT used in Section 5.5, we introduce a version with a vectorized state space, dubbed PARROT-state, since the method is originally only applicable in visual settings. The only modification consists in replacing the image encoder with a 3-layer MLP with 256 neurons per layer and ReLU activations. In this setting, the input to the encoder is therefore vector-based (non-visual).

**KL-regularization**  The baseline method introduced in Section 5.2 replaces the entropy term in SAC with a KL-regularization term encouraging the policy to math the learned prior, as done in Pertsch et al. (2020). Existing hyperparameters are inherited from SAC, and the prior is modeled as a Gaussian in order to compute KL-terms in closed form. The inverse of reward scale $\alpha$ was tuned in the range $[2e^{-4}, 2]$, and set to $2e^{-3}$.

### K.4 SFP

**Prior**  We model all families of state-free priors with conditional Real NVP Flows (Ardizzone et al., 2019), sharing the same architecture across all experiments. The invertible transformation $f_\theta$ is a composition of 6 coupling layers, each followed by a batch-normalization layer (Dinh et al., 2017). For each layer, a 3-layer MLP with 128 hidden units per layer is used to preprocess the conditioning input. Scale and transition networks are also implemented as a shared 3-layer MLP with 128 hidden units, whose output is split in two to provide scale and shift coefficients. All MLPs use ReLU activations. We train our state-free priors following the same protocol used with behavioral priors (100 epochs, a batch size of 400 samples and Adam (Kingma & Ba, 2015), with a learning rate of 0.0001, $\beta_1 = 0.9$, $\beta_2 = 0.999$ and a weight decay penalty of $1e-6$).

**Downstream Learning**  Our method does not introduce additional hyperparameters with respect to SAC, excluding those involved with the design and optimization of the neural network parameterizing the mixing function $\Lambda_\omega$. This is implemented as a 2-layer ReLU network with 128 units per layer and a sigmoid for output activation. We report that scaling the gradients for $\omega$ by a fixed coefficient $\epsilon$ was found beneficial for a stabler training of the mixing network ($\epsilon = 1e-9$ for `meta-world`, `point-maze` and $\epsilon = 1e-7$ for `gym-mujoco`). As exploration is mainly driven by the prior, SFP also relies on lower $\alpha$ parameters, namely $\alpha = 0.01$ for `meta-world`, `point-maze` and $\alpha = 0.005$ for `gym-mujoco`.

Finally, we introduce two implementation choices motivated by the prior knowledge that, during early stages of training, exploration should mainly be driven by samples from the action prior $\bar{\pi}$. First, while SAC samples action uniformly for the first 10000 steps to encourage exploration, SFP directly samples actions from its prior instead. Second, we initialize the bias parameter of the final layer of the mixing network $\Lambda_\omega$ to $\sigma^{-1}(\lambda_0)$ in order to target mixing weights around $\lambda_0 = 0.95$ during early training.

All remaining hyperparameters are shared with SAC (see Table 2).

**Test-time Policy**  At test time, instead of sampling an action from the mixture between prior $\bar{\pi}$ and policy $\pi$, the agent executes an action corresponding to the mean of the policy distribution. We note that this is consistent to what is done in SAC Haarnoja et al. (2018).

## L  Detailed Results

This section reports individual results were aggregated in previously occurring figures. In particular, Figure 18 and Figure 19 report performance for each `meta-world` and `point-maze` task, respectively. Figure 20 reports the evolution of average mixing weights as training progresses for all `meta-world` tasks (an aggregation can be found in Figure 14). Figures 21 and 22 also report performance for the two suites, but focus on visual RL settings. Figure 23 disentangles the results from Figure 9 across the five `mujoco-gym` environments.

For completeness, we also include plots for `meta-world` tasks that remain out of reach of all tested methods, namely `peg-insert-side`, `pick-place` and `push`. In general, these tasks require both deep exploration and precise grasping and control. While SFP remains capable of deep exploration, precise manipulation constitutes a challenge, as expert demonstrations do not contain grasping behavior (see Section 6). In visual settings, due to increased complexity, we observed a drop in performance on `reach`, `button-press` and `door-open` across all methods.

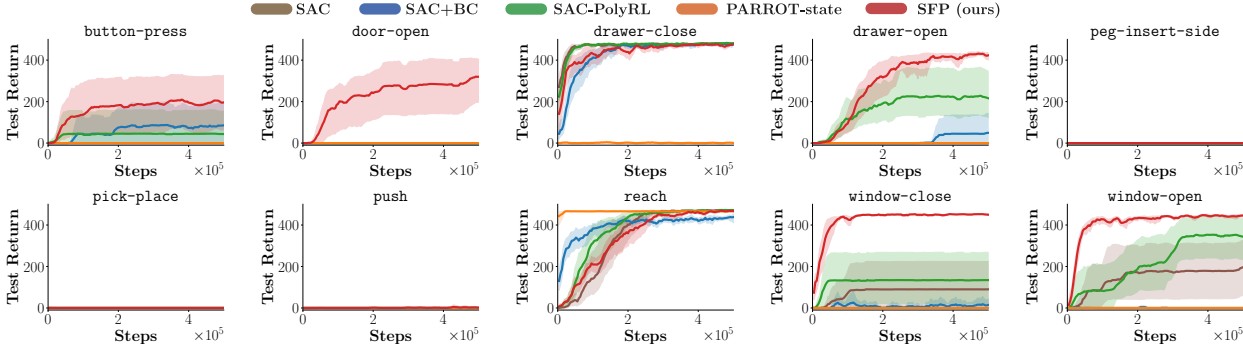

**Figure 18:** Learning curves for each downstream `meta-world` task.

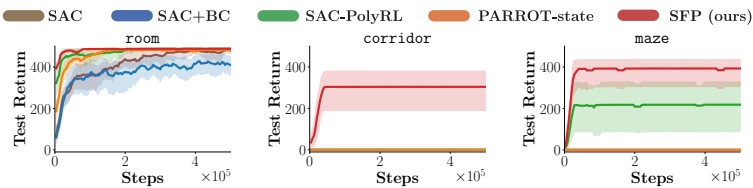

**Figure 19:** Learning curves for each downstream `point-maze` task.

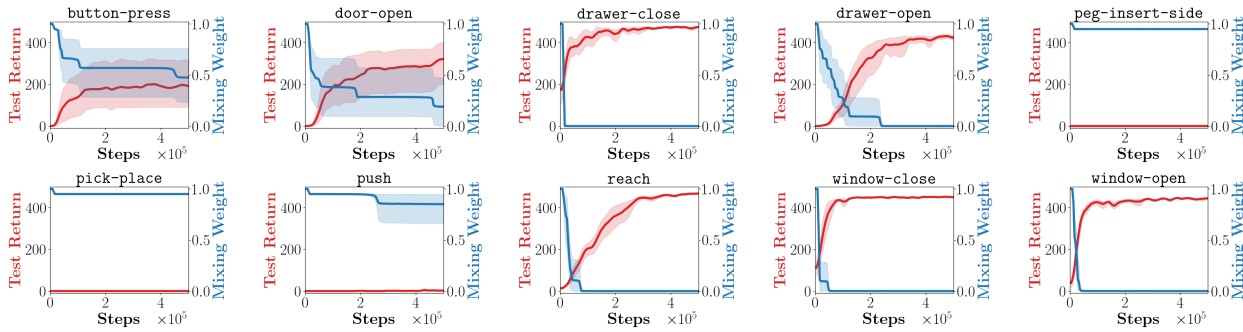

**Figure 20:** Evolution of mixing weight $\lambda$ and test performance for each downstream `meta-world` task.

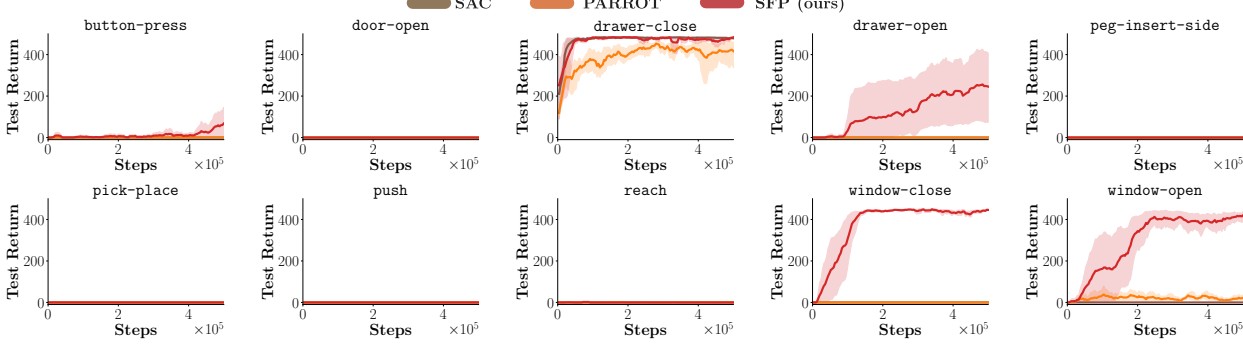

**Figure 21:** Learning curves for each downstream `meta-world` task in visual settings.

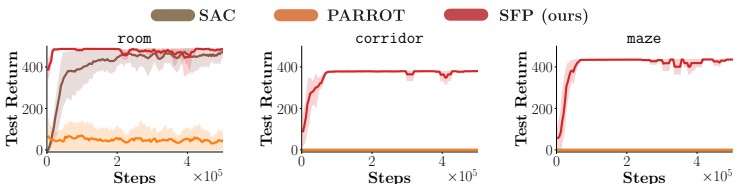

**Figure 22:** Learning curves for each downstream `point-maze` task in visual settings.

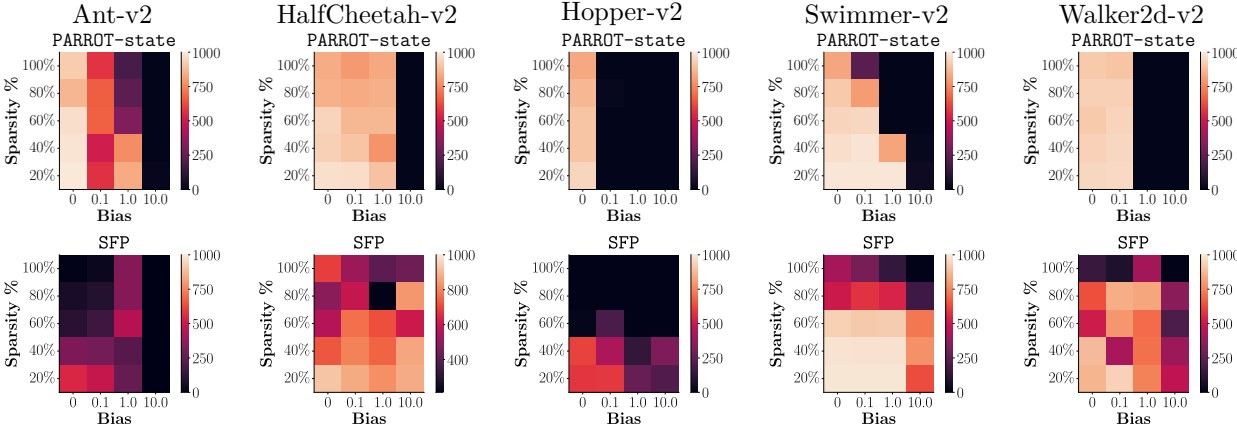

**Figure 23:** Mean performance on each `gym-mujoco` environment for different sparsity thresholds and bias norms.

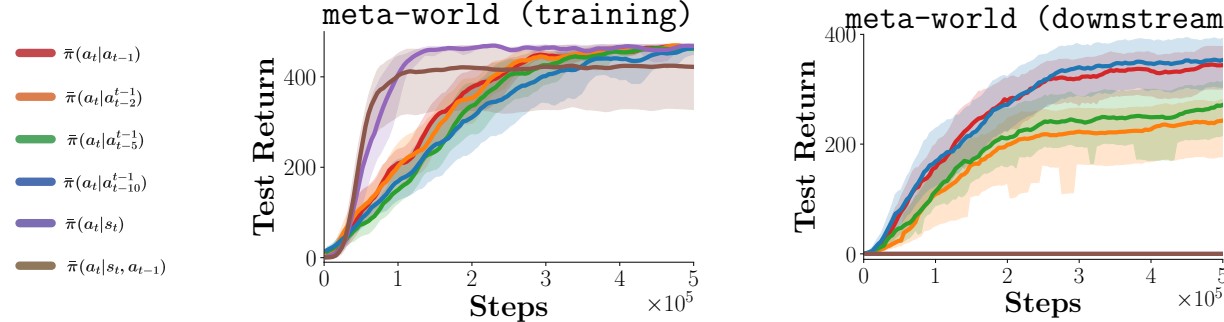

**Figure 24:** Comparison of downstream performance of action priors with different conditioning variables from Subsection 5.3, aggregated over downstream tasks via IQM.

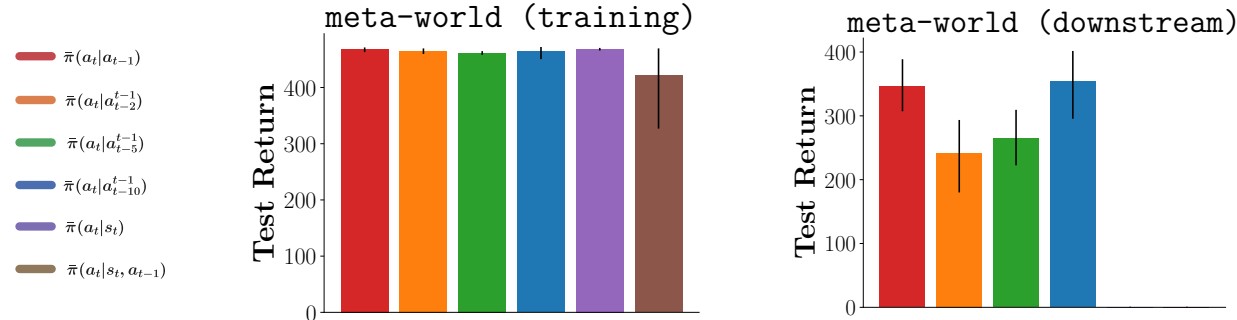

**Figure 25:** Comparison of final downstream performance of action priors with different conditioning variables from Subsection 5.3, aggregated over downstream tasks via IQM.

## M    Alternative Aggregation and Visualization

In this Section, we present an alternative aggregation scheme for key figures in the main paper. In particular, we compute the IQM with 95% stratified basic boostrap confidence interals as suggested in Agarwal et al. (2021). We further provide bar plots representing the final performance for each figure. In particular, Figures 24, 26, and 28 use IQM aggregation for the data used for Figures 4, 7, and 8 respectively. Figures 25, 27, and 29 report bar plots to represent the final IQM over task performances in Figures 4, 7, and 8 respectively.

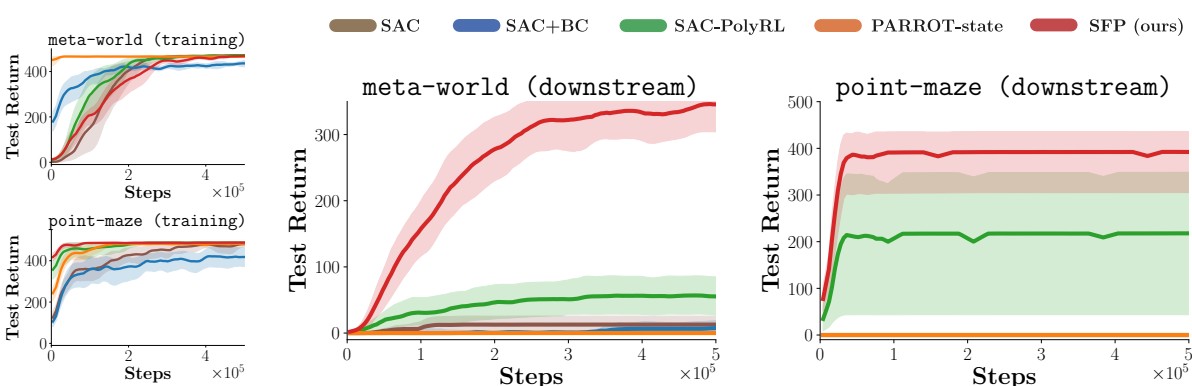

**Figure 26:** Comparison of SFP with several baselines from Subsection 5.4. Results are aggregated across tasks via IQM, instead of mean.

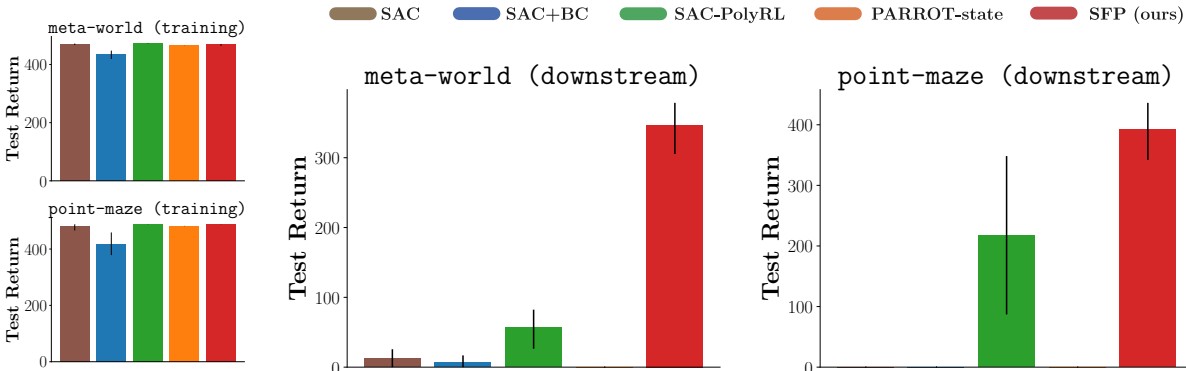

**Figure 27:** Comparison of SFP with several baselines from Subsection 5.4. Final returns are aggregated across tasks via IQM, instead of mean.

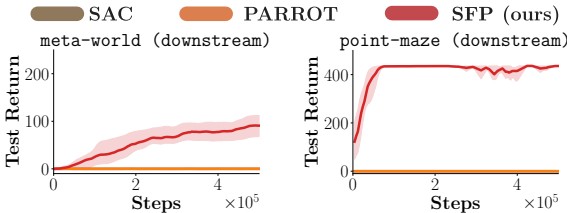

**Figure 28:** Performance on downstream learning in visual settings from Subsection 5.5, aggregated via IQM.

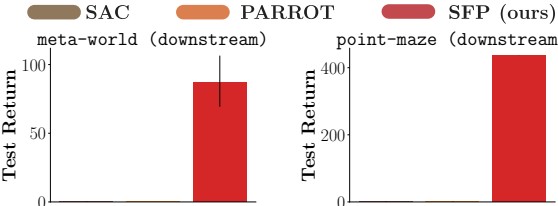

**Figure 29:** Final performance on downstream learning in visual settings from Subsection 5.5, aggregated via IQM.

