# OpenReview forum: "SFP: State-free Priors for Exploration in Off-Policy Reinforcement Learning"
_TMLR — Accepted by TMLR_

### Review · Reviewer_SWNm · 2022-06-18

**Summary Of Contributions:**

The authors introduce temporal priors, an extension of behavioral priors, which use provided expert trajectories to accelerate learning by guiding the policy towards relevant data/increasing coverage of the state space. Temporal priors are state-independent, and instead conditioned on the sequence of prior actions.

They motivate this difference by suggesting that when considering task transfer (i.e. expert trajectories are provided for task A but we are interested in task B), state-dependent priors are more likely to overfit to the provided behavior, limiting their effectiveness on new tasks.

Main contributions:
- Introduction of action-conditioned behavioral priors (temporal priors).
- Provide a way to integrate temporal priors into the formulation of SAC.
- Demonstrate the effectiveness of temporal priors to accelerate learning, particularly in the case of task transfer on the meta-world and point-maze benchmarks.

**Broader Impact Concerns:**

I have no concerns rearding the broader impact statement made by the authors.

**Requested Changes:**

- (High importance). The authors need to improve the presentation of the results. I would suggest replacing Figures (4, 6, 8, 9) with better aggregation or a different presentation of the results such as bar graphs (as done in Atari), tables, presenting a subset of learning curves, or taking in consideration some of the ideas/recommendations in [1]. The authors also need to clearly label if/what aggregation is occurring and details such as what the shaded area represents and the amount of expert data.
- (Mid importance). I would like to see [3] included as a baseline and ideally some methods which don't use behavioral priors but perform well on MetaWorld.
- (Mid importance). Additional justifications/clarity should be added to the places I noted. I would also like to see a bit more details in the body of the paper on the derivation of the objectives.
- (Mid-low importance). The paper would be improved with some additional ablations, particularly varying the amount of expert data used and changing the training task used for transfer, as [2] noted this can substantially affect performance.
- (Low importance). The biased downstream learning experiments seems a bit overfit to making PARROT perform poorly. I would be interested in seeing it repeated with bias/noise on the actions rather than the states.

References:
- [1] Agarwal, Rishabh, et al. "Deep reinforcement learning at the edge of the statistical precipice." 2021.
- [2] Singh, Avi, et al. "Parrot: Data-Driven Behavioral Priors for Reinforcement Learning." 2020.
- [3] Pertsch, Karl, Youngwoon Lee, and Joseph Lim. "Accelerating Reinforcement Learning with Learned Skill Priors." 2021.

**Strengths And Weaknesses:**

**Strengths:**

- The idea of removing the state-dependency in expert-provided trajectories is, as far as I'm aware a novel approach, which I'm surprised works as well as it does. This alone makes the work interesting in my opinion.
- Significant performance gap over baselines on provided tasks in the transfer (downstream) setting.
- High quality presentation/polish. Visually, the paper looks good, and the figures are easy to follow.
- Thorough appendix with details, analysis, etc. Code is provided & reproducibility feels high.

**Weaknesses:**

Experiments: The experiments part of the paper is very weak in my opinion.
- Presentation of results: While the results are visually pleasing, how the authors have chosen to present the results is at best confusing, and at worse deceptive. In Figures (4, 6, 8, 9) multiple tasks are aggregated into a single figure. This is particularly bad as the fact that this aggregation has occurred is not stated and there is not description of how it occurred (i.e mean? median?). There are shaded areas surrounding learning curves but there is no description of what they represent. See [1] for some possible discussion/suggestions on result aggregation. In general, there is a lot of information that is unavailable without combing through the appendix, such as the number of demonstrations used.
- Baselines/Environments: The baselines overall felt lacking. The authors make extensive reference to both [2,3] and yet only [2] is included as a baseline. Notably while [2] shows some negative examples for transfer, [3] focuses on transfer/downstream tasks, so it feels like a critical baseline. Given the performance of SAC-PolyRL, other exploration methods could also be useful as a strong baseline.

  In terms of environments, compared to prior work [2,3], the domains examined seem relatively simplistic. For example, on the training task, the performance is strong across all baselines, and in particular, PARROT seems to learn in <20k timesteps, suggesting these tasks may be too easy. Examining Figure 15, many tasks are solved almost immediately, or not at all.

  Part of the issue being that I did not expect any of the baselines to perform well. While I'm not super familiar with what is state-of-the-art on MetaWorld, [4] for example has some good results, including on "insert peg side" where all the baselines used in this paper fail (as well as TempoRL). Some baselines that can learn well on MetaWorld would allow for a better understanding of how much benefit, in terms of learning speedup, TempoRL provides.

Writing: There are a few places where the writing comes across as vague and omits necessary details to follow the logic the authors are presenting.
- "The main challenge introduced by temporal priors stems from their non-Markovianity, which renders existing integrations schemes unsuitable Singh et al. (2021); Tirumala et al. (2020)." This statement confused me because (1) these prior methods are not explained in the paper, so I'm not sure why they are necessarily not amenable to a non-Markovian prior and (2) it's unclear to me TempoRL deals with this problem either. As far as I can tell, the objectives Equation (3-6) ignore the non-Markovian nature of the prior.
- "the entropy term is computed on the policy alone, as it is not desirable to incentivize exploration of states according to the prior's entropy." I'm not sure why this is true. Additionally, since this change affects the derivation of SAC (as now the entropy is no longer over the current policy), I'm not sure why it's correct either.

Novelty: While I feel the authors do enough to make the work interesting, it is worth mentioning that the novelty isn't too high. The proposed prior is a small modification of behavioral priors and the integration into SAC is very similar to existing work in the learning from demonstrations field, particular Q-filter [5].

Typos:
- Missing period: "between the policy and temporal prior Crucially" (page 2).
- Should be \citep: "if they are rather different from those demonstrated Parisi et al. (2021)" (page 1).

References:
- [1] Agarwal, Rishabh, et al. "Deep reinforcement learning at the edge of the statistical precipice." 2021.
- [2] Singh, Avi, et al. "Parrot: Data-Driven Behavioral Priors for Reinforcement Learning." 2020.
- [3] Pertsch, Karl, Youngwoon Lee, and Joseph Lim. "Accelerating Reinforcement Learning with Learned Skill Priors." 2021.
- [4] Kumar, Aviral, Abhishek Gupta, and Sergey Levine. "Discor: Corrective feedback in reinforcement learning via distribution correction." 2020.
- [5] Nair, Ashvin, et al. "Overcoming exploration in reinforcement learning with demonstrations." 2018.

---

> ### Author Response · Authors · 2022-06-28
> **Response to Reviewer SWNm**
>
> We thank the reviewer for their insightful comments, particularly regarding the presentation of results and additional baselines. We address each concern individually below.
>
> **Comment:**
> *(High importance). The authors need to improve the presentation of the results. I would suggest replacing Figures (4, 6, 8, 9) with better aggregation or a different presentation of the results such as bar graphs (as done in Atari), tables, presenting a subset of learning curves, or taking in consideration some of the ideas/recommendations in [1]. The authors also need to clearly label if/what aggregation is occurring and details such as what the shaded area represents and the amount of expert data.*
>
> **Response:**
> We have redesigned the plots to address these concerns and reflect recommendations from [1]. The main design points are three: uncertainty quantification, graph type, and metrics.
> * For uncertainty quantification,  we recomputed shaded areas to report 95% stratified basic bootstrap confidence intervals, as suggested.
> * We have created bar plots for each learning curve in the main paper and added them to Appendix M. However, we would argue in favour of keeping learning curves in the main paper, as they are the standard representation for works interested in sample efficiency [2,3]. Only reporting final performance would obfuscate the number of steps required to converge. For this reason, we have added bar plots to Appendix M, and kept learning curves in the main paper. We would again ask for the reviewer’s opinion on this.
> * We have experimented with different metrics, namely mean, median and IQM. While IQM is suggested in [1], we would like to argue in favour of using a simple mean aggregation. Unlike in ATARI, our suites do not involve human normalization, and all tasks share the same range of achievable returns. For this reason, good performance on a single task does not have as large of an effect on the aggregated metric. Moreover, our suites have a smaller number of tasks compared to ATARI, thus reported metrics need to be statistically efficient [1] in order to provide informative results. For instance, using mean aggregation would still allow ranking the performance of baselines that solve less than 25% of the tasks, unlike IQM or median aggregation. For these reasons, we have not yet updated the main plots to use a different aggregation than the mean, but have added alternative visualisations of the most important plots using IQM in Appendix M. Similarly, the newly added bar plots also use IQM aggregation. We would like to ask the reviewer for their opinion on the choice of placement, and mean aggregation.
>
> As suggested, we clarified the current aggregation scheme in writing in the main body (p. 8, bottom), as well as what the shaded area represents and the amount of expert data used.
> We believe that this point was instrumental in improving the clarity of our presentation, and further feedback from the reviewer can help improve it further. We also remain open to any particular suggestion regarding individual plots.
>
> **References:**
> * [1] Agarwal, Rishabh et al. “Deep Reinforcement Learning at the Edge of the Statistical Precipice.” 2021
> * [2] Singh, Avi, et al. "Parrot: Data-Driven Behavioral Priors for Reinforcement Learning." 2020.
> * [3] Pertsch, Karl, Youngwoon Lee, and Joseph Lim. "Accelerating Reinforcement Learning with Learned Skill Priors." 2021.

---

> > ### Author Response · Authors · 2022-06-28
> > **Response to Reviewer SWNm - 2**
> >
> > **Comment:**
> > *(Mid importance). Additional justifications/clarity should be added to the places I noted. I would also like to see a bit more details in the body of the paper on the derivation of the objectives.*
> >
> > **Response:**
> > More details on intermediate derivation steps were added in p.7, top. We report the detailed comments as follows, and address them individually:
> >
> > *Baselines/Environments: The baselines overall felt lacking. The authors make extensive reference to both [2,3] and yet only [2] is included as a baseline. Notably while [2] shows some negative examples for transfer, [3] focuses on transfer/downstream tasks, so it feels like a critical baseline. Given the performance of SAC-PolyRL, other exploration methods could also be useful as a strong baseline.
> > In terms of environments, compared to prior work [2,3], the domains examined seem relatively simplistic. For example, on the training task, the performance is strong across all baselines, and in particular, PARROT seems to learn in <20k timesteps, suggesting these tasks may be too easy. Examining Figure 15, many tasks are solved almost immediately, or not at all.
> > Part of the issue being that I did not expect any of the baselines to perform well. While I'm not super familiar with what is state-of-the-art on MetaWorld, [4] for example has some good results, including on "insert peg side" where all the baselines used in this paper fail (as well as TempoRL). Some baselines that can learn well on MetaWorld would allow for a better understanding of how much benefit, in terms of learning speedup, TempoRL provides.*
> >
> > We added an empirical comparison to SPIRL [3] in Appendix H. We agree that this method is designed for transfer to downstream tasks. However, crucially, SPIRL still requires tasks to have a state distribution which matches the distribution of demonstrations trajectories. For instance, in maze experiments from [3], the global structure of the maze changes, but the agent only has access to a local view (a square centered on it), and as a result the distributions of states in different maze types largely overlap. On the other hand, in our setting, state distributions can vary drastically from the training task to the downstream task. As such, SPIRL runs into the same limitation that PARROT faces. We empirically validated this hypothesis by training SPiRL on metaworld. We observed that the performance of SPiRL in downstream tasks matches PARROT-state’s performance, and falls short of TempoRL. Performance on the training task is however slightly reduced with respect to PARROT-state, due to the fact that the low-level policy is trained to imitate training data, which do not contain stationary behavior. As a consequence, SPiRL does not have the option to keep the end effector static, and can only achieve suboptimal behavior in the reaching task.
> >
> > As for the concern for simplistic domains, we note that the tasks that PARROT can quickly solve are those for which the policy used for collecting the data is enough for reaching strong performance. In this case, PARROT leverages what can be seen as expert demonstration for the task, and we believe that its performance is consistent with this advantage.
> >
> > Finally, concerning further baselines, we note that most papers evaluating on metaworld use dense rewards [4], and evaluate mainly multi-task or meta-learning. For instance, [5] evaluates both the single-task and the multi-task setup, but relies on dense reward functions. On the other hand, we train our policy on each task separately, and use sparse reward to benchmark exploration strategies. This training protocol is largely unexplored. For instance, within single-task learning, the original paper reports that SAC performs well with dense rewards (Fig. 11 in [4]). However, we observed that, in sparse settings, SAC performance falls drastically. In other words, baselines that perform well in multi-learning, dense reward metaworld tasks are not necessarily strong contenders in our single-task, sparse rewards setup. For this reason, we report baselines that we found to be performing reasonably within our setting of interest, independently from their performance in multi-task, dense rewards settings.
> >
> > [this response continues in the following comment]
> >
> > **References:**
> > * [2] Singh, Avi, et al. "Parrot: Data-Driven Behavioral Priors for Reinforcement Learning." 2020.
> > * [3] Pertsch, Karl, Youngwoon Lee, and Joseph Lim. "Accelerating Reinforcement Learning with Learned Skill Priors." 2021.
> > * [4] Yu, Tianhe et al. “Meta-World: A Benchmark and Evaluation for Multi-Task and Meta Reinforcement Learning.” 2019
> > * [5] Kumar, Aviral et al. "Discor: Corrective Feedback in Reinforcement Learning via Distribution Correction." 2020.

---

> > > ### Author Response · Authors · 2022-06-28
> > > **Response to Reviewer SWNm - 3**
> > >
> > > *"The main challenge introduced by temporal priors stems from their non-Markovianity, which renders existing integrations schemes unsuitable Singh et al. (2021); Tirumala et al. (2020)." This statement confused me because (1) these prior methods are not explained in the paper, so I'm not sure why they are necessarily not amenable to a non-Markovian prior and (2) it's unclear to me TempoRL deals with this problem either. As far as I can tell, the objectives Equation (3-6) ignore the non-Markovian nature of the prior.*
> > >
> > > We now justify this statement and provide empirical validation in an additional experimental Section (5.2), where we compare our integration scheme against action warping and KL-regularization. In the case of action space warping [2], the prior is effectively integrated in the environment dynamics: as a result, if the prior is not conditioned on the past state alone, the environment loses its Markov property, and stationary policies are no longer guaranteed to be optimal. On the other hand, penalizing the KL-divergence between policy and a non-Markovian prior [6, 3] encourages the state-conditioned policy to match the distribution of a potentially non-stationary prior, which is an ill-posed objective. Instead, our mixture-based integration only suffers from biased learning objectives in the case of a non-Markovian action prior (see Appendix A). Empirically, we found this to be a mild limitation, as our integration achieves sensibly better performance compared to the two baselines in both training and downstream tasks.
> > >
> > > *"the entropy term is computed on the policy alone, as it is not desirable to incentivize exploration of states according to the prior's entropy." I'm not sure why this is true. Additionally, since this change affects the derivation of SAC (as now the entropy is no longer over the current policy), I'm not sure why it's correct either.*
> > >
> > > While the epistemic uncertainty in the prior’s output could also be used to guide exploration, we choose this formulation (i.e. computing entropy on the policy alone) for two reasons. First, the entropy of the mixture can be maximized by a uniform policy, and by setting the mixing coefficients to 0 (i.e. only sampling actions from the policy). Hence, optimizing over the entropy of the mixture could favor solutions which disregard the prior. A second, more practical consideration is that this particular formulation does not require evaluating the likelihood of actions under the prior, which could be impractical without having full access to a probabilistic generative model. We would kindly ask the reviewer to clarify any remaining concern on correctness.
> > >
> > > *Novelty: While I feel the authors do enough to make the work interesting, it is worth mentioning that the novelty isn't too high. The proposed prior is a small modification of behavioral priors and the integration into SAC is very similar to existing work in the learning from demonstrations field, particular Q-filter [5].*
> > >
> > > We thank the reviewer for pointing out Q-filter, which we now acknowledge explicitly when discussing the learning rule for the mixing parameter (p. 7). While similar in spirit, we remark that Q-filter intuitively casts the advantage of demonstrated actions as a binary signal for modulating a behavioral cloning loss, while we backpropagate it in order to fit a mixture between prior and policy, and derive it directly from a principled objective.
> > > As the reviewer points out, our proposed family of priors can be seen as a modification to behavioral priors. However, to the best of our knowledge, a thorough empirical evaluation, and a suitable integration scheme for non-Markovian priors are not present in existing literature.
> > >
> > > Finally, we thank the reviewer for pointing out typos, which have now been corrected.
> > >
> > > **References:**
> > > * [2] Singh, Avi, et al. "Parrot: Data-Driven Behavioral Priors for Reinforcement Learning." 2020.
> > > * [3] Pertsch, Karl, Youngwoon Lee, and Joseph Lim. "Accelerating Reinforcement Learning with Learned Skill Priors." 2021.
> > > * [6] Tirumala, Dhruva et al. “Behavior Priors for Efficient Reinforcement Learning”, 2020

---

> > > > ### Author Response · Authors · 2022-06-28
> > > > **Response to Reviewer SWNm - 4**
> > > >
> > > > **Comment:**
> > > > *(Mid-low importance). The paper would be improved with some additional ablations, particularly varying the amount of expert data used and changing the training task used for transfer, as [2] noted this can substantially affect performance.*
> > > >
> > > > **Response:**
> > > > We point the reviewer to Appendix C for an ablation on the amount and quality of expert data. In particular, we observe that our method does not suffer strongly from reduced amounts of training data On the other hand, reducing the data quality (i.e. substituting uniformly sampled actions in the training data) has a more drastic effect, and performance significantly drops when substituting 50% or more of actions, as this alters local temporal structures. The second concern is instead addressed in the newly introduced Appendix D, in which we report results for a different, more informative training task (pick-place). A summary of results can be found in the General Comment.
> > > >
> > > > **Comment:**
> > > > *(Low importance). The biased downstream learning experiments seems a bit overfit to making PARROT perform poorly. I would be interested in seeing it repeated with bias/noise on the actions rather than the states.*
> > > >
> > > > **Response:**
> > > > The setting of biased downstream learning was chosen because it does not impact the performance of the naive RL algorithm (e.g. vanilla SAC), but can be problematic for prior-based methods. This allows us to only measure the performance degradation due to the prior design itself.
> > > > We set out to repeat the experiments, replacing the bias of the state space with some form of corruption of the action space. In particular, we reverse the action vector before executing it in the environment (e.g. action=action[::-1]). We choose this transformation due to its simplicity, and because it does not alter the difficulty of the environment for a naive RL algorithm (e.g. vanilla SAC), unlike injecting noise in the action space. We evaluate this setup empirically in Appendix E. In short, we observe that, under this transformation, both TempoRL and PARROT-state observe a performance drop. In the case of temporal priors, this can be partially traced back to the fact that they may be conditioned on OOD actions. Most importantly, for both temporal and behavioral prior, this transformation disrupts actions suggested by the prior, as the trajectories it was trained on could not be reproduced in the downstream environment.

---

### Review · Reviewer_J9Aw · 2022-06-19

**Summary Of Contributions:**

The paper explores the transfer of behavior priors, learned on an offline dataset of diverse experience, to accelerate reinforcement learning on a new downstream task. It specifically investigates scenarios in which the behavior priors are solely conditioned on past actions p(a_t | a_t-1) instead of the environment state p(a_t | s_t), as is commonly done in prior work. In experimental evaluations on transfer problems, including Metaworld, the paper shows that non-action conditioned priors transfer better to unseen state spaces and allow transfer from state-based pre-training data to visual RL downstream tasks.

**Broader Impact Concerns:**

The provided broader impact statement seems sufficient to me.

**Requested Changes:**

See "weaknesses" section above, particularly points A-D seem critical for acceptance to me.

**Strengths And Weaknesses:**

## Strengths
The use of behavior priors can help to solve one of the notorious problems of (deep) RL: sample efficiency. The core problem the paper tackles -- learning more generalizable behavior priors -- is important for wider applicability of behavior priors and thus impactful in the community.

The paper is well-written and easy to follow. It briefly summarizes the relevant background on behavior priors in RL and then clearly lays out the proposed idea(s). The figures help to communicate the ideas throughout the paper. The experimental evaluations highlight the scenarios in which the proposed priors lead to superior generalization and give a good intuition *why* they lead to faster learning downstream with qualitative examples.

I like the "additional applications" section (5.4) since the authors show some less intuitive but relevant use cases for the proposed prior design -- transferring priors from state to visual observations and transferring in the presence of systematic corruptions in training or downstream task observations.

I also found the second contribution of the paper interesting: apart from the state-free prior design, the paper proposes an alternative to the usual KL-regularization approach for using priors to guide downstream learning: they use directly learn the policy as a weighted mixture of the prior and a learned policy, vaguely similar to a residual RL formulation with a fixed "base policy" (the learned behavior prior) and a residual policy that is trained on the downstream task, just that instead of adding the two (as I believe is commonly done in residual RL), they mix them probabilistically and sample from one or the other during downstream learning based on a learned advantage estimate. This way of using the learned behavior prior is novel to my knowledge and can be impactful even independent of the particular state-free prior design proposed in the paper.


## Weaknesses

While I like the general direction of the paper and the presentation of the overall idea, there are a few weaknesses that should be fixed before I can recommend acceptance.

(A) **More detailed discussion of prior work on information asymmetry**: The concept that withholding certain information / inputs from learned behavior priors will lead to better generalization (at the cost of lower prior expressivity) has been well-discussed in prior work (eg [1-3]). Tirumala et al [2] explicitly mention on page 8 of their paper the option to condition behavior priors only on prior actions instead of states. Although all of these papers focus their experimental evaluations on different information asymmetries than this paper (eg training prior on only proprioceptive state information), the fact that information asymmetry is not a novel concept introduced in this paper should be adequately discussed in the related work. I believe the main novelty of this paper with regards to the prior design is the actual experimental evaluation of the state-free priors which was not performed in prior works. The writing should reflect this.

(B) **Cleanly distinguish between two contributions in experimental evaluation**: The paper makes two contributions: the state-free behavior prior design and the novel approach to using the prior to accelerate downstream learning. However, the paper focusses all the attention in the experimental evaluation on the first point and conflates the contribution of the second point. Instead, both contributions should be cleanly investigated separately. Eg the paper could first establish that state-free priors lead to superior transfer (with the existing experiments) and then ablate the choice of prior regularization technique, comparing to the two most common approaches: (1) softly regularizing the policy towards the prior with a KL constraint, (2) warping the action space like in PARROT. Specifically the first evaluation is missing from the paper now, but is vital to show the benefits of the proposed policy mixing scheme.

(C) **Show performance decrease for downstream tasks with in-distribution states**: It is intuitive that a state-free prior will generalize better to OOD states in the downstream task. But it is equally intuitive that the learned prior will be less expressive than a state-conditioned prior, since it is essentially blind to the agent's current state. It would be important to quantify this trade-off by showing the performance of the proposed prior design in scenarios where the downstream task has states that are in-distribution to the training data & compare to state-conditioned priors. The paper already has such evaluations, but only on the same tasks as used for collecting the training data. In contrast, prior works that use behavior priors for accelerating RL have shown transfer to *unseen* tasks with *seen* state distributions. It seems important to show evaluations of the state-free vs state-conditioned priors in this scenario -- eg evaluations on the commonly used FrankaKitchen environment could be added to show the performance decline one would need to accept for in-distribution state scenarios (this could serve to motivate future research into automatically figuring out the best prior conditioning).

(D) **Naming of "temporal priors" not ideal**: The authors call their proposed state-free prior design "temporal priors" and their method accordingly TempoRL, presumably since temporal correlations between actions is all a state-free prior can learn to model. However, prior works have investigated conditioning on prior temporal information (like past states) too, eg [3]. The *temporal* aspect of the proposed priors is not the novel part, but the *state-free* part. Thus, I suggest the authors consider renaming their approach to focus on the state-free aspect of the learned priors rather than the temporal modeling aspect to clearly distinguish it from other temporal prior designs.


## Minor points and suggestions

There were a few minor points, which I don't see as critical to the acceptance of the paper, but could improve the overall quality of the paper:

- It could be nice to give an intuitive example for a scenario in which the structure of the observations would not be known while learning the prior (and thus a state-free prior is the best modeling decision) -- ie it is clear that state-free priors reduce the required assumptions on the data, but the importance of this could be made clearer with an example.

- Regarding the sentence in section 3.1: "... we do not require high quality trajectories to be collected on a task that is similar to the one at hand." This has two parts: (1) high quality, (2) on a related task to the target task -- it would be good to make clearer which of these is not required (presumably (2), while (1) is still required to learn a reasonable prior).

- It would be interesting to show the development of the learned mixing coefficient lambda over time / some visualizations of situations where it is high vs low to get better insight into what it learns.

- The characterization in the conclusion that state-conditioned priors are superior when "the data collection policy can solve the downstream task" is too narrow -- they should work well when the downstream state spaces are in distribution of the training state space, even if new tasks need to be learned.

- It could be interesting to apply the proposed priors to the CALVIN environment: CALVIN provides large unstructured datasets that can be used for training behavior priors and crucially features four different environment layouts. One could expect that state-conditioned priors would have trouble generalizing between the environments, while the proposed state-free priors could generalize much better.

## Questions
- I am surprised to see in Fig 4 that p(a | s, a_t-1) is doing better than p(a | s) --> it should suffer from the same domain shift -- do you have any intuition why it is doing better? --> It would also be really interesting to see this comparison in an environment where state-based transfer should work (see point (C) above) to see whether conditioning on past actions too is generally useful in these scenarios too.

- How hard is it to accurately learn the advantage used to update the lambda parameter? I have experimented with advantage weighting for prior regularization myself before (although in the context of KL-regularized RL) and found it very hard to accurately learn to estimate the advantage, leading to meaningless regularization weights. Did you find any particularly sensitive parameter in that optimization?



[1] Galashov et al., Information asymmetry in kl-regularized RL, 2019

[2] Tirumala et al., Behavior priors for efficient reinforcement learning, 2020

[3] Salter et al., Priors, Hierarchy, and Information Asymmetry for Skill Transfer in Reinforcement Learning, 2020

---

> ### Author Response · Authors · 2022-06-28
> **Response to Reviewer J9Aw**
>
> We thank the reviewer for the detailed suggestions and their in-depth review. We particularly appreciate the suggestions regarding the experimental evaluation, which have contributed to improving the presentation of our results. We address all comments in this response, and provide a revision of the paper including suggested changes and additional experimental results.
>
> **Comment:**
> *A) More detailed discussion of prior work on information asymmetry*
>
> **Response:**
> We thank the reviewer for highlighting the additional prior works, and for suggesting a more detailed positioning of our method’s novelty. We accordingly addressed this in the Related Works section (p.3, bottom), which were extended with a discussion on existing empirical studies on information asymmetry. We remain open to further elaborate on this topic.
>
> **Comment:**
> *(B) Cleanly distinguish between two contributions in experimental evaluation*
>
> **Response:**
> We understand the importance of the suggested evaluation, and therefore we added Subsection 5.2, which includes new experiments involving existing integration methods (i.e. KL-regularization and action warping) applied to “temporal” priors (i.e. $\bar\pi(a_{t}|a_{t-1})$). Empirically, this subsection shows how existing integration methods suffer when coupled with state-free priors, while our proposed mixture-based integration achieves better performance within this setting.
> In particular, we restructured the experimental section as follows: after motivating the design of a state-free prior in Subsection 5.1, we directly move on to validating the new integration schemes in Subsection 5.2, as suggested by the reviewer. The experimental section then proceeds as before, by highlighting state space coverage (5.3) and extensive results on transfer learning (5.4).
>
> **Comment:**
> *(C) Show performance decrease for downstream tasks with in-distribution states*
>
> **Response:**
> We understand that this is an important point, and that the current discussion of limitations can be extended through an empirical evaluation. For this reason, we introduce a new ablation evaluating state-free priors and behavioral priors in unseen tasks with seen state distributions (Appendix D). In particular, in order to avoid hyperparameter tuning on a new suite, we simply choose pick-place as our new training task. This allows us to evaluate transfer to unseen tasks with in-distribution states (push), and to evaluate temporal priors’ limitations in expressiveness. A summary of the results is presented in the General Comment.
>
> **Comment:**
> *(D) Naming of "temporal priors" not ideal*
>
> **Response:**
> We are open to renaming our paper to “State-free Priors for Exploration in Off-Policy Reinforcement Learning”, and accordingly change the name of the method, as well as all occurrences of “temporal prior”. We will make this change if the other reviewers also accept the proposal.
>
> **Comment:**
> *It could be nice to give an intuitive example for a scenario in which the structure of the observations would not be known while learning the prior (and thus a state-free prior is the best modeling decision) -- ie it is clear that state-free priors reduce the required assumptions on the data, but the importance of this could be made clearer with an example.*
>
> **Response:**
> We add a motivating example in the introduction (p. 1, bottom), which should help provide some early intuition on potential applications. In particular, we considered the scenario of a robot that was trained from scratch in a simple environment (e.g. it learned to walk in a flat room), and needs to be retrained to traverse a complex construction site. Even if observations in the second environments are drastically different from those previously received (e.g. due to different lighting conditions and terrain patterns), relevant information could still be recovered from past experiences. We argue that this could be addressed through state-free priors.
>
> **Comment:**
> *Regarding the sentence in section 3.1: "... we do not require high quality trajectories to be collected on a task that is similar to the one at hand." This has two parts: (1) high quality, (2) on a related task to the target task -- it would be good to make clearer which of these is not required (presumably (2), while (1) is still required to learn a reasonable prior).*
>
> **Response:**
> The most important restriction that we lift is that training trajectories do not have to be collected on a task that is very similar to the downstream task (i.e. (2)). We thank the reviewer for the pointer, and we reformulated the sentence on p. 4 to clarify this.

---

> > ### Author Response · Authors · 2022-06-28
> > **Response to Reviewer J9Aw - 2**
> >
> > **Comment:**
> > *It would be interesting to show the development of the learned mixing coefficient lambda over time / some visualizations of situations where it is high vs low to get better insight into what it learns.*
> >
> > **Response:**
> > We direct the reviewer to Appendix F, in which we already present a discussion and an aggregated visualization for the development of the mixing coefficients. For the updated version, we now also include per-task mixing coefficient plots in Appendix L.
> >
> > **Comment:**
> > *The characterization in the conclusion that state-conditioned priors are superior when "the data collection policy can solve the downstream task" is too narrow -- they should work well when the downstream state spaces are in distribution of the training state space, even if new tasks need to be learned.*
> >
> > **Response:**
> > We agree with the remark, and reworded the conclusion to reflect this (p. 12).
> >
> > **Comment:**
> > *It could be interesting to apply the proposed priors to the CALVIN environment: CALVIN provides large unstructured datasets that can be used for training behavior priors and crucially features four different environment layouts. One could expect that state-conditioned priors would have trouble generalizing between the environments, while the proposed state-free priors could generalize much better.*
> >
> > **Response:**
> > We appreciate the suggestion, and believe that this environment can be an interesting benchmark even for methods which do not leverage its NLP components. Due to time constraints, and considering how this was marked as a minor comment, we are unable to provide results on this environment in the context of this rebuttal. We hope that the significant overlap with reported experiments (ie. the newly reported ablation on a more informative task in Appendix D) can compensate for this.
> >
> > **Question:**
> > *I am surprised to see in Fig 4 that p(a | s, a_t-1) is doing better than p(a | s) --> it should suffer from the same domain shift -- do you have any intuition why it is doing better? --> It would also be really interesting to see this comparison in an environment where state-based transfer should work (see point (C) above) to see whether conditioning on past actions too is generally useful in these scenarios too.*
> >
> > **Response:**
> > We further experiment on this by training action-state conditional priors on pick-place trajectories, and deploying them on downstream metaworld tasks. We report results in Appendix D. In this case, we observe that action-state conditional priors underperforms with respect to state-conditional priors, but their performance still falls between that of purely state-conditional and action-conditional priors (i.e. $p(a_t|s_t)$ and $p(a_t|a_{t-1})$) respectively. This suggests that state-action conditional priors offer a “middle ground” between temporal and behavioral priors. Intuitively, in the original setting in Fig. 4, we hypothesize that the slight increase in performance can be traced back to the fact that action-state conditional prior can in principle focus on their action input, and be less reliant on their state input, which can be beneficial under shifts in the state distribution.
> >
> > **Question:**
> > *How hard is it to accurately learn the advantage used to update the lambda parameter? I have experimented with advantage weighting for prior regularization myself before (although in the context of KL-regularized RL) and found it very hard to accurately learn to estimate the advantage, leading to meaningless regularization weights. Did you find any particularly sensitive parameter in that optimization?*
> >
> > **Response:**
> > We also found that learning the mixing coefficient $\lambda$ was not straightforward, and required strong regularization. In particular, we found the following two settings to be beneficial:
> > * We initialize the bias in the last layer of the mixing network $\Lambda$ in order to output large mixing weights at the beginning of training, making the assumption that the prior will be useful in early stages.
> > * A low learning rate is adopted to stabilize learning, and prevent noisy gradients from resulting in large parameter updates in a single optimization step.

---

> > > ### Comment · Reviewer_J9Aw · 2022-07-05
> > > **Thanks for the comprehensive rebuttal! -- Post-rebuttal feedback**
> > >
> > > Thank you for the rebuttal updates. The rebuttal addresses most of the points raised in my review. Below are more detailed post-rebuttal comments:
> > >
> > > - (A) related work is now adequately covered.
> > >
> > > - (B) experiments now distinguish between two contributions: the added section shows the comparison I asked for, thanks! The result could be made even more useful by either describing how the alternative integration methods were tuned (eg how many KL regularization coefficients were tried) or, even better, if results on a domain could be provided where the alternative integration approaches were tuned by their authors (eg FrankaKitchen), since they would present a stronger baseline.
> > >
> > > - (C) thanks for evaluating the in-distribution state case. The results in Section D seem important since they provide additional context for the results in the experimental section and show the limitations of the state-free priors, so they should be discussed more prominently in the main experimental section, especially also with regard to more complex, object-interaction tasks.
> > >
> > > - (D) the suggested renaming sounds good to me
> > >
> > > - for the SPiRL baseline in section H it would be good to also try lower regularization weights (or higher target divergences) to allow the policy to deviate further from the prior and better learn the target task -- usually the regularization strength needs to be retuned for a new task and simply copying the parameter from FrankaKitchen is unlikely to result in good performance.

---

> > > > ### Author Response · Authors · 2022-07-06
> > > > **Response to Reviewer J9Aw**
> > > >
> > > > We thank the reviewer for considering our comments, and for their additional answer.
> > > >
> > > > We understand that, according to submission guidelines, the rebuttal window has closed. Thus, we kindly ask the action editor to clarify if we are allowed to address the additional comments in a further rebuttal answer, or whether we should wait for the decision, and include the additional experiments in the final version in case of acceptance.

---

> > > > > ### Comment · Action_Editors · 2022-07-08
> > > > > **You may send a response to the AE**
> > > > >
> > > > > Dear authors, the discussion period is indeed over. At this stage, if you have any last responses, you may leave a brief clarificatory message for my benefit as I compile the reviews, but I would request you not to continue any conversations with reviewers.

---

### Review · Reviewer_7Mab · 2022-06-21

**Summary Of Contributions:**

The paper presents a technique for incorporating behavioral priors in reinforcement learning, where a behavioral prior is a model over actions learned using some previously collected (near-expert) data. In contrast to prior works which typically learn a state-conditioned prior, this paper learns a temporal prior, wherein the action selected by the prior at any given time step depends only on the prior action(s) taken in that environment. This requires the prior dataset to contain temporally directed behavior, but the state space for the prior and downstream tasks can be different. The authors combine their learned temporal prior with an existing off-policy RL algorithm (SAC) in a straightforward way: the exploration policy in SAC becomes a mixture of two policies: the learned temporal prior, and the policy learned by SAC. The mixing weight for this mixture policy is itself learned via the maximum entropy objective in SAC. The authors show results on a suite of simulated manipulation and navigation tasks, and find that temporal priors hold significant advantage when state distribution is different for training tasks and downstream tasks.


**Broader Impact Concerns:**

I do not have any broader impacts concern for this work.

**Requested Changes:**

- The authors talk a lot about their approach being “task-agnostic”, but I don’t think any kind of data-driven exploration method is truly task agnostic - you need some kind of shared structure between the tasks in the offline dataset (training task) and the new tasks you are trying to learn (downstream task). There is more nuanced discussion about this in Section 4.1, so I believe some of it should be noted in the introduction as well (i.e. avoid overpromising to readers).
- Related work. Paper is missing an important citation in the “temporally extended exploration” section of the paper. Bogdanovic et al [1] do something very similar, except they condition on the history of past states, as opposed to the last action. For the kind of trajectories considered in this paper (a robot end effector reaching a specific point), I accept the practical behavior of these learned priors to be similar (but the prior work requires shared state spaces).
- Visual RL results. While the proposed technique does better than the baseline in the visual RL setting, it is important to note that TempoRL fails to make progress on a majority visual RL tasks, as shown in FIgure 17 in the Appendix (but is still better than the baseline which does not work on any visual tasks in this setting). This should be noted in the main paper, and some discussion should be provided on why this is the case.
I would encourage the authors to open-source their code for this project, as it would help accelerate research in this area.

[1] Miroslav Bogdanovic and Ludovic Righetti. ​​Learning to Explore in Motion and Interaction Tasks. IEEE International Workshop on Intelligent Robots and Systems (IROS), 2019. https://arxiv.org/abs/1908.03731


**Strengths And Weaknesses:**

### Strengths
- Simple, easy-to-implement approach
- Can be useful in a number of settings where there is a state-space mismatch between the training tasks and the downstream tasks

### Weaknesses
- As is often the case in this line of work, the learned priors are not very helpful when the downstream tasks require substantially novel behavior. For example, since the priors in this work are mainly trained on reaching tasks, they cannot help pick and place objects in downstream tasks (as acknowledged by the authors, and as seen in Figure 15 in the appendix).

---

> ### Author Response · Authors · 2022-06-28
> **Response to Reviewer 7Mab**
>
> We thank the reviewer for the clear review, and for their comments on related works and clarity. We are happy to introduce the requested changes in the paper, and remain available to address further concerns.
>
> **Comment:**
> *The authors talk a lot about their approach being “task-agnostic”, but I don’t think any kind of data-driven exploration method is truly task agnostic - you need some kind of shared structure between the tasks in the offline dataset (training task) and the new tasks you are trying to learn (downstream task). There is more nuanced discussion about this in Section 4.1, so I believe some of it should be noted in the introduction as well (i.e. avoid overpromising to readers).*
>
> **Response:**
> We understand this concern, as we use the term  ‘task-agnostic’ loosely throughout the paper. We have addressed this by adding a paragraph in the introduction clarifying our assumptions on training trajectories (p. 2). In case this is deemed insufficient, we are open to replacing occurrences of ‘task-agnostic’ with a less nuanced word, such as ‘weakly informative’.
>
> **Comment:**
> *Related work. Paper is missing an important citation in the “temporally extended exploration” section of the paper. Bogdanovic et al [1] do something very similar, except they condition on the history of past states, as opposed to the last action. For the kind of trajectories considered in this paper (a robot end effector reaching a specific point), I accept the practical behavior of these learned priors to be similar (but the prior work requires shared state spaces).*
>
> **Response:**
> We thank the reviewer for the pointer to this paper. We agree that it is very relevant to our approach and have integrated it into related works, highlighting similarities and differences (p. 3).
>
> **Comment:**
> *Visual RL results. While the proposed technique does better than the baseline in the visual RL setting, it is important to note that TempoRL fails to make progress on a majority visual RL tasks, as shown in FIgure 17 in the Appendix (but is still better than the baseline which does not work on any visual tasks in this setting). This should be noted in the main paper, and some discussion should be provided on why this is the case. I would encourage the authors to open-source their code for this project, as it would help accelerate research in this area.*
>
> **Response:**
> We believe that the drop in performance in visual settings can be traced back to the inherent complexity of learning in a state space with a much larger dimension (by approximately ~300x in our case) [1]. We added this discussion in the main paper (p. 12, top), and described the performance in the visual settings more clearly. The code for the project has already been made available on the project website (see footnote on page 3), and will be updated to include the experiments requested in the rebuttal.
>
> **References:**
> * [1] Srinivas, Aravind et al. “CURL: Contrastive Unsupervised Representations for Reinforcement Learning.” 2020

---

> > ### Comment · Reviewer_7Mab · 2022-07-04
> > **Thanks for changes; comments on "task-agnostic"**
> >
> > Thank for your changes and response.
> >
> > I feel the following statement does not quite convey the requirements regarding shared structure between train/test tasks: " While such trajectories should display generally desirable qualities, such as correlation and directedness, they are not required to contain any information that is specific to the task at hand, and are therefore referred to as task-agnostic in the context of this paper."
> >
> > For example, I can have directed trajectories that always get the robot to raise its end-effector as high as possible, but that won't be useful when trying to learn a new task in which you need to interact with objects placed on a table in front of you. As a result, I would suggest adding clearer statements regarding the requirements for some shared structure between train/test for the method to work, avoid vague phrases like "generally desirable", and removing "task-agnostic", as suggested above.

---

> > > ### Author Response · Authors · 2022-07-04
> > > **Response to Reviewer 7Mab**
> > >
> > > We thank the reviewer for considering our comments, and for their further suggestions. As suggested, we have thus reformulated the sentence as follows:
> > >
> > > *While we assume that such trajectories display qualities such as correlation and directdeness, and in particular reflect good exploration strategies for the task at hand,  we do not require them to explicitly encode state-action dependencies specific to downstream tasks.*
> > >
> > > Moreover, we have removed the occurrence of 'task-agnostic' in this sentence, and replaced other occurrences with 'weakly informative'.

---

> > > > ### Comment · Reviewer_7Mab · 2022-07-05
> > > > **Thank you**
> > > >
> > > > Thanks for taking my suggestions into account. I do not have any further comments.

---

### Author Response · Authors · 2022-06-28
**General Comment**

We thank all reviewers for their constructive feedback. We have addressed each review individually, and will use this comment to summarize the most important changes and additions.


**Section 5.2**

We added a new experimental section, which empirically compares our mixture-based integration for temporal priors, with integrations based on KL-regularization [1], and on flow-based transformation of the action space [2]. Our experiments suggest that our proposed integration scheme is better suited to temporal priors with respect to existing techniques.

**Training Task Ablation**

We added an ablation study for the choice of training task as Appendix D. In particular, we  select a more complex training task for the meta-world suite, namely pick-place. We then report downstream performance over all meta-world tasks for both temporal and behavioral priors. We highlight results for unseen tasks that can be considered in-distribution (e.g. the push task is very similar with respect to the training task, with the exception that the final goal is on the table). We also discuss how the structure of training demonstrations affect performance. As expected, we find behavioral prior to outperform temporal priors in the training task, as well as for in-distribution tasks. Furthermore, behavioral priors trained on these informative trajectories can solve tasks that were unsolved when trained on simple reaching trajectories. On the other hand, temporal priors suffer due to lack of expressiveness in the most complex tasks (e.g. push), including the training task (i.e. pick-place). When also considering remaining downstream tasks, temporal priors however close the performance gap, as they still outperform behavioral priors on tasks which are fundamentally different from the training task. A more detailed discussion is provided in Appendix D.

**Appendix E, H**

We added a comparison with an additional baseline (SPIRL [2]) in Appendix H, and a further ablation for mujoco experiments in Appendix E.

**Updated Plots**

We updated all plots to show 95% stratified basic bootstrap confidence intervals through error bars or shaded intervals (originally, they showed standard deviation across seeds).

**Updated Related Works**

We incorporated and discussed all references suggested by reviewers in Section 2.

**References:**
* [1] Singh, Avi, et al. "Parrot: Data-Driven Behavioral Priors for Reinforcement Learning." 2020.
* [2] Pertsch, Karl, Youngwoon Lee, and Joseph Lim. "Accelerating Reinforcement Learning with Learned Skill Priors." 2021.

---

### Decision · Action_Editors · 2022-08-07

**Recommendation:** Accept with minor revision

**Comment:**

This paper proposes a novel and surprisingly simple approach to distill offline data from an environment into "priors" that can usefully guide exploration even in new environments and new tasks. The key idea is simply to omit states like prior methods have done, and instead model action-only sequences. Through evaluation that has become significantly more robust and thorough over the course of the rebuttal discussions, they show that this does indeed perform well, particularly for transfer to new environments/new state spaces, where prior methods would be operating out of domain and hence perform poorly. The new experiments added during the rebuttal phase more clearly circumscribe where this method is useful, pointing out limitations.

A second interesting idea is also proposed for probabilistically combining policy prescriptions from the prior policy and a new task-specific policy trained with SAC, and is now evaluated to show gains over alternative approaches for influencing task policies with priors.

Connections to prior work have also been more clearly stated now. I thank the reviewers and authors for a very productive review and rebuttal phase.

As the paper stands, it is nearly ready for publication. Most comments from reviewers have already been addressed in the paper. As reviewer J9Ab suggests, I recommend that the authors change the name of the method (and correspondingly, the title) to something more transparent and indicative of the key idea: "state-free", or "action-only" behavioral priors. I recommend acceptance for publication after this minor revision. If the authors so please, they may acknowledge the anonymous reviewers for their contributions towards improving the manuscript. No other changes necessary.